# HOW DOES OVERPARAMETRIZATION AFFECT PERFORMANCE ON MINORITY GROUPS?

## ABSTRACT

The benefits of overparameterization for the overall performance of modern machine learning (ML) models are well known. However, the effect of overparameterization at a more granular level of data subgroups is less understood. Recent empirical studies demonstrate encouraging results: (i) when groups are not known, overparameterized models trained with empirical risk minimization (ERM) perform better on minority groups; (ii) when groups are known, ERM on data subsampled to equalize group sizes yields state-of-the-art worst-group-accuracy in the overparameterized regime. In this paper, we complement these empirical studies with a theoretical investigation of the risk of overparameterized random feature models on minority groups. In a setting in which the regression functions for the majority and minority groups are different, we show that overparameterization always improves minority group performance.

## 1 INTRODUCTION

Traditionally, the goal of machine learning (ML) is to optimize the *average* or *overall* performance of ML models. The relentless pursuit of this goal eventually led to the development of deep neural networks, which achieve state of the art performance in many application areas. A prominent trend in the development of such modern ML models is overparameterization: the models are so complex that they are capable of perfectly interpolating the data. There is a large body of work showing overparameterization improves the performance of ML models in a variety of settings (*e.g.* ridgeless least squares Hastie et al. (2019), random feature models Belkin et al. (2019); Mei and Montanari (2019b) and deep neural networks Nakkiran et al. (2019)).

However, as ML models find their way into high-stakes decision-making processes, other aspects of their performance (besides average performance) are coming under scrutiny. One aspect that is particularly relevant to the fairness and safety of ML models is their performance on traditionally disadvantaged demographic groups. There is a troubling line of work showing ML models that perform well on average may perform poorly on minority groups of training examples. For example, Buolamwini and Gebru (2018) show that commercial gender classification systems, despite achieving low classification error on average, tend to misclassify dark-skinned people. In the same spirit, Wilson et al. (2019) show that pedestrian detection models, despite performing admirably on average, have trouble recognizing dark-skinned pedestrians.

The literature examines the effect of model size on the generalization error of the worst group. Sagawa et al. (2020) find that increasing model size beyond the threshold of zero training error can have a negative impact on test error for minority groups because the model learns spurious correlations. They show that subsampling the majority groups is far more successful than upweighting the minority groups in reducing worst-group error. Pham et al. (2021) conduct more extensive experiments to investigate the influence of model size on worst-group error under various neural network architecture and model parameter initialization configurations. They discover that increasing model size either improves or does not harm the worst-group test performance across all settings. Idrissi et al. (2021) recommend using simple methods, *i.e.* subsampling and reweighting for balanced classes or balanced groups, before venturing into more complicated procedures. They suggest that newly developed robust optimization approaches for worst-group error control (Sagawa et al., 2019; Liu et al., 2021) could be computationally demanding, and that there is no strong (statistically significant) evidence of advantage over those simple methods.

In this paper, we provide theoretical justification for the empirical results in Sagawa et al. (2020); Pham et al. (2021); Idrissi et al. (2021) by studying how overparameterization affects the performance of ML models on minority groups in an idealized regression problem. Our investigation shows that *overparameterization generally improves or stabilizes the performance of ML models on minority groups*. Our main contributions are:

1. we develop a simple two-group model for studying the effects of overparameterization on (sub)groups. This model has parameters controlling signal strength, majority group fraction, overparameterization ratio, discrepancy between the two groups, and error term variance that display a rich set of possible effects.

2. we develop a comprehensive picture of the limiting risk of empirical risk minimization in a high-dimensional asymptotic setting (see Sections 3).

3. we show that majority group subsampling provably improves minority group performance in the overparameterized regime.

Some of the technical tools that we develop in the proofs may be of independent interest.

## 2 PROBLEM SETUP

### 2.1 DATA GENERATING PROCESS

Let $\mathcal{X} \subset \mathbf{R}^d$ be the feature space and $\mathcal{Y} \subset \mathbf{R}$ be the output space. To keep things simple, we consider a two group setup. Let $P_0$ and $P_1$ be probability distributions on $\mathcal{X} \times \mathcal{Y}$. We consider $P_0$ and $P_1$ as the distribution of samples from the minority and majority groups respectively. In the minority group, the samples $(x, y) \in \mathcal{X} \times \mathcal{Y}$ are distributed as

$$x \sim P_X, \ \ y \mid x = \beta_0^\top x + \varepsilon, \ \ \varepsilon \sim N(0, \tau^2), \tag{2.1}$$

where $P_X$ is the marginal distribution of the features, $\beta_0 \in \mathbf{R}^d$ is a vector of regression coefficients, and $\tau^2 > 0$ is the noise level. The normality of the error term in (2.1) is not important; our theoretical results remain valid even if the error term is non-Gaussian.

In the majority group, the marginal distribution of features is identical, but the conditional distribution of the output is different:

$$y \mid x = \beta_1^\top x + \varepsilon, \ \ \varepsilon \sim N(0, \tau^2), \tag{2.2}$$

where $\beta_1 \in \mathbf{R}^d$ is a vector of regression coefficients *for the majority group*. We note that this difference between the majority and minority groups is a form of *concept drift* or *posterior drift*: the marginal distribution of the feature is identical, but the conditional distribution of the output is different. We focus on this setting because it not only simplifies our derivations, but also isolates the effects of concept drift between subpopulations through the difference $\delta \triangleq \beta_1 - \beta_0$. If the covariate distributions between the two groups are different, then an overparameterized model may be able to distinguish between the two groups, thus effectively modeling the groups separately. In that sense, by assuming that the covariates are equally distributed, we consider the worst case.

Let $g_i \in \{0, 1\}$ denote the group membership of the $i$-th training sample. The training data $\{(x_i, y_i, g_i)\}_{i=1}^n$ consists of a mixture of samples from the majority and minority groups:

$$\begin{aligned} g_i &\sim \mathsf{Ber}(\pi) \\ (x_i, y_i) \mid g_i &\sim P_{g_i} \end{aligned}, \tag{2.3}$$

where $\pi \in [\frac{1}{2}, 1]$ is the (expected) proportion of samples from the majority group in the training data. We denote $n_1$ as the sample size for majority group in the training data.

### 2.2 RANDOM FEATURE MODELS

Here, we consider a random feature regression model (Rahimi and Recht, 2007; Montanari et al., 2020)

$$f(x, a, \Theta) = \sum_{j=1}^N a_j \sigma(\theta_j^\top x / \sqrt{d}) \tag{2.4}$$

where $\sigma(\cdot)$ is a non-linear activation function and $N$ is the number of random features considered in the model. The random feature model is similar to a two-layer neural network, but the weights $\theta_j$'s of the hidden layer are set at some (usually random) initial values. In other words, a random feature model fits a linear regression model on the response $y$ using the non-linear random features $\sigma(\theta_j^\top x/\sqrt{d})$'s instead of the original features $x_j$'s. This type of models has recently been used Montanari et al. (2020) to provide theoretical explanation for some of the behaviors seen in large neural network models.

We note that at first we analyzed the usual linear regression model hoping to understand the effect of overparameterization. Specifically, we considered the model

$$f(x, a) = a^\top x. \tag{2.5}$$

However, we found that the linear model suffers in terms of prediction accuracy for minority group in overparameterized regime. In particular, in the case of high signal-to-noise ratio, the prediction accuracy for the minority group worsens with the overparameterization of the model. This is rather unsatisfactory as the theoretical findings do not echo the empirical findings in Le Pham et al. (2021), suggesting that a linear model is not adequate for understanding the deep learning phenomena in this case. We point the readers to Appendix A for the theoretical analysis of overparameterezied linear models.

## 2.3 TRAINING OF THE RANDOM FEATURE MODEL

We consider two ways in which a learner may fit the random feature model: empirical risk minimization (ERM) that does not require group annotations; subsampling that does require group annotations. The availability of group annotations is highly application dependant and there is a plethora of prior works considering either of the scenarios (Hashimoto et al., 2018; Zhai et al., 2021; Pham et al., 2021; Sagawa et al., 2019; 2020; Idrissi et al., 2021).

### 2.3.1 EMPIRICAL RISK MINIMIZATION (ERM)

The most common way to fit a predictive model is to minimize the empirical risk on the training data:

$$\hat{a} \in \underset{a \in \mathbf{R}^N}{\arg\min} \frac{1}{n} \sum_{i=1}^n \frac{1}{2} \{y_i - f(x_i, a, \Theta)\}^2 = \underset{a \in \mathbf{R}^N}{\arg\min} \frac{1}{n} \sum_{i=1}^n \frac{1}{2} \{y_i - \sum_{j=1}^N a_j \sigma(\theta_j^\top x_i/\sqrt{d})\}^2, \tag{2.6}$$

where we recall that the weights $\theta_j$'s of the first layer are randomly assigned. The above optimization (2.6) has a unique solution when the number of random features or neurons are less than the sample size ($N < n$) and we call this regime as *underparameterized*. When the number of neurons is greater than the sample size ($N > n$), which we call the *overparameterized* regime, $\hat{a}$ is not unique in (2.6), as there are multiple $a \in \mathbf{R}^N$ that interpolate the training data, *i.e.*, $y_i = \sum_{j=1}^N a_j \sigma(\theta_j^\top x_i)$, $i \in [n]$, resulting in a zero training error. In such a situation we set $\hat{a}$ at a specific $a \in \mathbf{R}^N$ which (1) interpolates the training data and (2) has minimum $\ell_2$-norm. This particular solution is known as the *minimum norm interpolant solution* (Hastie et al., 2019; Montanari et al., 2020) and is formally defined as

$$\hat{a} \in \arg\min\{\|a\|_2 : y_i = \sum_{j=1}^N a_j \sigma(\theta_j^\top x_i/\sqrt{d}), \ i \in [n]\} = (Z^\top Z)^\dagger Z^\top y, \tag{2.7}$$

where $Z \in \mathbf{R}^{n \times N}$ with $Z_{i,j} = \sigma(\theta_j^\top x_i/\sqrt{d})$ and $A^\dagger$ is the Moore-Penrose inverse of $A$. The minimum norm interpolant solution (2.7) has an alternative interpretation: it is the limiting solution to the ridge regression problem at vanishing regularization strength,

$$\hat{a} = \lim_{\lambda \to 0+} \hat{a}_\lambda, \ \hat{a}_\lambda \in \underset{a \in \mathbf{R}^N}{\arg\min} \left[ \frac{1}{n} \sum_{i=1}^n \frac{1}{2} \{y_i - \sum_{j=1}^N a_j \sigma(\theta_j^\top x_i/\sqrt{d})\}^2 + \frac{N\lambda}{d} \right]. \tag{2.8}$$

In fact the conclusion in (2.8) continues to hold in the underparameterized ($N < n$) regime. Hence, we combine the two regimes ($N < n$ and $N > n$) and obtain the ERM solution as (2.8). In Hastie et al. (2019) it is known as the *ridgeless* solution. Finally, having an estimator $\hat{a}$ of the parameter $a$ in the random feature model (2.4), the response of an individual with feature vector $x$ is predicted as:

$$\hat{y}(x) = f(x, \hat{a}, \Theta) = \sum_{j=1}^N \hat{a}_j \sigma(\theta_j^\top x/\sqrt{d}). \tag{2.9}$$

### 2.3.2 MAJORITY GROUP SUBSAMPLING

It is known that a model trained by ERM often exhibits poor performance on minority groups (Sagawa et al., 2019). One promising way to circumvent the issue is *majority group subsampling*: a randomly drawn subset of training sample points are discarded from majority groups to match the sample sizes for the majority and minority groups in the remaining training data. This emulates the effect of reweighted-ERM, where the samples from minority groups are upweighted in the ERM.

In underparameterized case the reweighting is preferred over the subsampling due to its superior statistical efficiency. On contrary, in the overparameterized case, reweighing may not have the intended effect on the performance of minority groups (Byrd and Lipton, 2019), but subsampling does (Sagawa et al., 2020; Idrissi et al., 2021). Thus we consider subsampling as a way of improving performance in minority groups in overparameterized regime. We note that subsampling requires the knowledge of the groups/sensitive attributes, so its applicability is limited to problems in which the group identities are observed in the training data.

## 3 RANDOM FEATURE REGRESSION MODEL

In this paper we mainly focus on the minority group performance, which we measure as the mean squared prediction error for the minority samples in the test data:

$$R_0(\hat{a}) = \mathbb{E}_{x \sim P_X}[\{f(x, \hat{a}, \Theta) - \beta_0^\top x\}^2] = \mathbb{E}_{x \sim P_X}[\{\sum_{j=1}^N \hat{a}_j \sigma(\theta_j^\top x/\sqrt{d}) - \beta_0^\top x\}^2], \quad (3.1)$$

where $x$ is independent of the training data, and is identically distributed as $x_i$'s and for a generic $a$ by the notation $\mathbb{E}_a$ we mean that the expectation is taken over the randomness in $a$. One may alternatively study $\mathbb{E}_{(x,y) \sim P_0}[\{f(x, \hat{a}, \Theta) - y\}^2]$ as the mean square prediction error for the minority group, but for a fixed noise level $\tau$ it has the same behavior as (3.1), as suggested by the following: $\mathbb{E}_{(x,y) \sim P_0}[\{f(x, \hat{a}, \Theta) - y\}^2] = \mathbb{E}_{x \sim P_X}[\{f(x, \hat{a}, \Theta) - \beta_0^\top x\}^2] + \tau^2$.

A finite sample study of $R_0(\hat{a})$ (3.1) is difficult, in this paper we study it's limiting behavior in high-dimensional asymptotic setup. Specifically, recalling the notations from Table 1 we consider $N/d \to \psi_1 > 0$, $n/d \to \psi_2 > 0$ and $n_1/n \to \pi \in [1/2, 1]$. The problem is underparameterized if $\gamma = \psi_1/\psi_2 = \lim_{d \to \infty}\{N/n\} < 1$ and overparameterized if $\gamma > 1$.

To study the minority risk we first decompose it into several parts and study each of them separately. The first decomposition follows:

**Lemma 3.1.** *We define the bias* $\mathscr{B}(\beta_0, \delta, \tau) = \mathbb{E}_{x \sim P_X}[\{\mathbb{E}_\epsilon[f(x, \hat{a}, \Theta)] - \beta_0^\top x\}^2]$ *and the variance* $\mathscr{V}(\beta_0, \delta, \tau) = \mathbb{E}_{x \sim P_X}[\text{Var}_\epsilon\{f(x, \hat{a}, \Theta)\}]$, *where* $\epsilon = (\epsilon_1, \ldots, \epsilon_n)^\top$ *are the errors in the data generating model, i.e.* $\epsilon_i = y_i - x_i^\top \beta_{g_i}$ *and* $\text{Var}_\epsilon$ *denotes the variance with respect to* $\epsilon$. *The minority risk decomposes as*

**Table 1:** Table of notations

| Notations | Descriptions |
|-----------|--------------|
| $n$ | Total sample size |
| $n_0, n_1$ | Sample-sizes in minority and majority groups |
| $N$ | Number of neurons |
| $d$ | Covariate dimension |
| $\psi_1$ | $N/d$ |
| $\psi_2$ | $n/d$ |
| $\gamma$ | Overparameterization $N/n$ |
| $\pi$ | Majority group proportion |
| $\beta_0, \beta_1$ | Regression coefficients for minority and majority groups |
| $\delta$ | $\beta_1 - \beta_0$ |
| $\tau^2$ | Noise variance |

$$\mathbb{E}_{x \sim P_X, \epsilon}[\{f(x, \hat{a}, \Theta) - \beta_0^\top x\}^2]$$
$$= \mathscr{B}(\beta_0, \delta, \tau) + \mathscr{V}(\beta_0, \delta, \tau). \quad (3.2)$$

The proof is realized by decomposing the square in $[f(x, \hat{a}, \Theta) - \beta_0^\top x]^2 = [\{f(x, \hat{a}, \Theta)\} - \mathbb{E}_\epsilon[f(x, \hat{a}, \Theta)]\} + \{\mathbb{E}_\epsilon[f(x, \hat{a}, \Theta)] - \beta_0^\top x\}]^2$. We now provide an interpretation for the variance term.

**The variance term** Firstly we notice that the variance $\mathscr{V}(\beta_0, \delta, \tau)$ captures the average variance of the prediction rule $\hat{f}(x, \hat{a}, \Theta)$ in terms of the errors $\epsilon$ in training data. To look closely we define

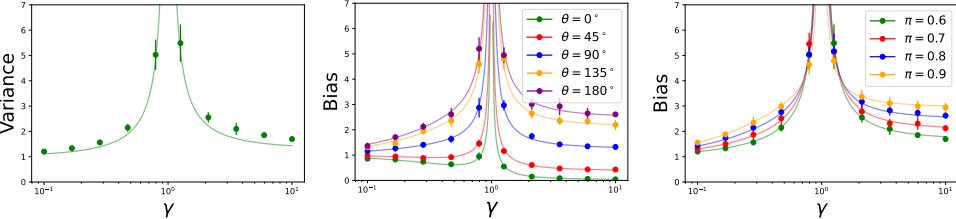

**Figure 1:** Trends for bias and variance in the minority ERM risk for varying overparameterization $\gamma$. *Left:* variances for varying $\gamma$. *Middle:* biases for different $\gamma$ and $\theta$, where $\theta$ is the angle between $\beta_0$ and $\beta_1$. Here the majority group proportion $\pi$ is set at 0.8. *Right:* biases for different $\pi$'s and $\gamma$'s when $\theta$ is set at $180°$. The solid lines are the theoretical predictions and the points with error-bars represent the empirical predictions over 10 random runs.

$z = \sigma(\Theta x/\sqrt{d})$ as the random feature for a new covariates $x$ and realize that the variance of the predictor with respect to $\epsilon$, and thus $\mathscr{V}(\beta_0, \delta, \tau)$, do not depend on the values of $\beta_0$ or $\delta$, as shown in the following display:

$$\text{Var}_\epsilon\{f(x, \hat{a}, \Theta)\} = \text{Var}_\epsilon\{z^\top \hat{a}\} = \text{Var}_\epsilon\{z^\top (Z^\top Z)^\dagger Z^\top y\} = \text{Var}_\epsilon\{z^\top (Z^\top Z)^\dagger Z^\top \epsilon\}.$$

Hence we drop $\beta_0$ and $\delta$ from the notation and write the variance term as $\mathscr{V}(\tau)$. Since $\epsilon_i$'s have variance $\tau^2$ we realize that the exact dependence of the variance term with respect to $\tau$ is the following: $\mathscr{V}(\tau) = \tau^2 \mathscr{V}(\tau = 1)$. This also implies that the variance term $\mathscr{V}(\tau)$ *teases out the contribution of data noise level $\tau$* in the minority risk.

**The bias term** We notice that the bias term $\mathscr{B}(\beta_0, \delta, \tau)$ in decomposition (3.2) is the average bias in prediction for the minority group. We further notice that it does not depend on the noise level $\tau$ (both $\mathbb{E}_\epsilon[f(x, \hat{a}, \Theta)]$ and $\beta_0^\top x$ do not depend on $\tau$) and denote it as $\mathscr{B}(\beta_0, \delta)$ by dropping $\tau$. Next, we separate out the contributions of $\beta_0$ and $\delta$ in the bias term via a decomposition. For the purpose we recall, $Z = \{z_{i,j}\}_{i \in [n], j \in [N]} \in \mathbf{R}^{n \times N}$ is the matrix of random features and define the following (1) $X = [x_1^\top, \dots x_n^\top]^\top$ is the covariate matrix in training data, (2) $X_1$ is the covariate matrix consisting only the samples from majority group, and (3) $Z_1$ is the random feature matrix for the majority group. The decomposition for the bias term follows:

$$\mathscr{B}(\beta_0, \delta) = \mathbb{E}_{x \sim P_X}[\{(z^\top (Z^\top Z)^\dagger Z^\top X - x^\top)\beta_0\}^2 + \{z^\top (Z^\top Z)^\dagger Z_1^\top X_1 \delta\}^2 \\ + 2(z^\top (Z^\top Z)^\dagger Z^\top X - x^\top)\beta_0 \delta^\top X_1^\top Z_1 (Z^\top Z)^\dagger z]. \tag{3.3}$$

Next, we make some assumptions on $\beta_0$ and $\delta$ vectors:

**Assumption 3.2.** *There exists some $F_\beta, F_\delta > 0$ and $F_{\beta, \delta} \in \mathbf{R}$ such that the following holds:* $\|\beta_0\|_2^2 = F_\beta^2$, $\|\delta\|_2^2 = F_\delta^2$ *and* $\langle \beta_0, \delta \rangle = F_{\beta, \delta}$.

Here, $F_\beta$ and $F_\delta$ are the $\ell_2$ signal strengths of $\beta_0$ and $\delta$. The decomposition in the next lemma separates out the contributions of $\beta_0$ and $\delta$ in $\mathscr{B}(\beta_0, \delta)$.

**Lemma 3.3.** *Define the followings:* $\mathscr{B}_\beta = \mathbb{E}_{x \sim P_X}[\|z^\top (Z^\top Z)^\dagger Z^\top X - x^\top\|_2^2/d]$, $\mathscr{B}_\delta = \mathbb{E}_{x \sim P_X}[\|z^\top (Z^\top Z)^\dagger Z_1^\top X_1\|_2^2/d]$ *and* $\mathscr{C}_{\beta, \delta} = 2\mathbb{E}_{x \sim P_X}[(z^\top (Z^\top Z)^\dagger Z^\top X - x^\top)X_1^\top Z_1 (Z^\top Z)^\dagger z]$. *Then we have* $\mathscr{B}(\beta_0, \delta) = F_\beta^2 \mathscr{B}_\beta + F_\delta^2 \mathscr{B}_\delta + F_{\beta, \delta} \mathscr{C}_{\beta, \delta}$.

In Lemma 3.3 we see that $F_\delta^2 \mathscr{B}_\delta + F_{\beta, \delta} \mathscr{C}_{\beta, \delta}$ quantifies the contribution of the model misspecification in the two-group model, and we call this *misspecification error term*. For the other terms in the decomposition we utilize the results of Montanari et al. (2020) who studied the effect of overparameterization on the overall performance (*i.e.*, in a single group model), thus the misspecification error term does not appear in their analysis. Our theoretical contribution in this paper is studying the *exact asymptotics of the misspecification error terms*. Before we present the asymptotic results we introduce some assumptions and definitions. First we make a distributional assumption on the covariates $x_i$'s and the random weights $\theta_j$'s which allow an easier analysis for the minority risk.

**Assumption 3.4.** *We assume that $\{x_i\}_{i=1}^n$ and $\{\theta_j\}_{j=1}^N$ are* iid $\text{Unif}\{\mathbb{S}^{d-1}(\sqrt{d})\}$, *i.e., uniformly over the surface of a $d$-dimensional Euclidean ball of radius $\sqrt{d}$ and centered at the origin.*

Next we assume that activation function $\sigma$ has some properties, which are satisfied by any commonly used activation functions, *e.g.* ReLU and sigmoid activations.

**Assumption 3.5.** *The activation function $\sigma : \mathbf{R} \to \mathbf{R}$ is weakly differentiable with weak derivative $\sigma'$ and for some constants $c_0, c_1 > 0$ it holds $|\sigma(u)|, |\sigma'(u)| \leq c_0 e^{c_1|u|}$.*

Both assumptions 3.4 and 3.5 also appear in Montanari et al. (2020). Below we define some quantities which we require to describe the asymptotic results.

**Definition 3.6.** *1. For the constants*

$$\mu_0 = \mathbb{E}\{\sigma(G)\}, \quad \mu_1 = \mathbb{E}\{G\sigma(G)\}, \quad \mu_\star^2 = \mathbb{E}\{\sigma(G)^2\} - \mu_0^2 - \mu_1^2, \qquad (3.4)$$

*where the expectation is taken with respect to $G \sim \mathrm{N}(0,1)$ we assume that $0 < \mu_0, \mu_1, \mu_\star < \infty$ and define $\xi = \mu_1/\mu_\star$.*

2. *Recall that $N/d \to \psi_1 > 0$, $n/d \to \psi_2 > 0$ and $n_1/n \to \pi \in [1/2, 1]$. We set $\psi = \min\{\psi_1, \psi_2\}$ and define*

$$\chi = -\tfrac{[(\psi\xi^2 - \xi^2 - 1)^2 + 4\xi^2\psi]^{1/2} + (\psi\xi^2 - \xi^2 - 1)}{2\xi^2} \ .$$

3. *We furthermore define the following:*

$$\begin{aligned}
\mathscr{E}_0^\star &= -\chi^5\xi^6 + 3\chi^4\xi^4 + (\psi_1\psi_2 - \psi_1 - \psi_2 + 1)\chi^3\xi^6 - 2\chi^3\xi^4 - 3\chi^3\xi^2 \\
&\quad + (\psi_1 + \psi_2 - 3\psi_1\psi_2 + 1)\chi^2\xi^4 + 2\chi^2\xi^2 + \chi^2 + 3\psi_1\psi_2\chi\xi^2 - \psi_1\psi_2, \\
\mathscr{E}_1^\star &= \psi_2\chi^3\xi^4 - \psi_2\chi^2\xi^2 + \psi_1\psi_2\chi\xi^2 - \psi_1\psi_2, \\
\mathscr{E}_2^\star &= \chi^5\xi^6 - 3\chi^4\xi^4 + (\psi_1 - 1)\chi^3\xi^6 + 2\chi^3\xi^4 + 3\chi^3\xi^2 + (-\psi_1 - 1)\chi^2\xi^4 - 2\chi^2\xi^2 - \chi^2 \ .
\end{aligned}$$

Equipped with the assumptions and the definitions we're now ready to state the asymptotics for each of the terms in the minority group prediction errors. The following lemma, which states the asymptotic results for $\mathscr{B}_\beta$ and $\mathscr{V}(\tau)$, has been proven in Montanari et al. (2020).

**Lemma 3.7** (Theorem 5.7, Montanari et al. (2020)). *Let the assumptions 3.2, 3.4 and 3.5 hold. Define $\mathscr{B}^\star = \mathscr{E}_1^\star / \mathscr{E}_0^\star$ and $\mathscr{V}^\star = \mathscr{E}_2^\star / \mathscr{E}_0^\star$ where $\mathscr{E}_0^\star$, $\mathscr{E}_1^\star$ and $\mathscr{E}_2^\star$ are defined in Definition 3.6. Then the following hold:*

$$\lim_{d\to\infty} \mathbb{E}[\mathscr{B}_\beta] = \mathscr{B}^\star, \ \ \lim_{d\to\infty} \mathbb{E}[\mathscr{V}(\tau)] = \tau^2 \mathscr{V}^\star \ ,$$

*where the expectation $\mathbb{E}$ is taken over $\{x_i\}_{i=1}^n$, $\{\theta_j\}_{j=1}^N$, $\beta_0$ and $\delta$.*

**Trend in the variance term** We again recall that the variance term $\mathscr{V}(\tau)$ teases out the contributions of noise $\epsilon$ in the minority group prediction error. Here, the trend that we're most interested in is the effect of overparameterization: how does the asymptotic $\mathscr{V}^\star$ behave as a function of $\gamma = \psi_1/\psi_2 = \lim_{d\to\infty}(N/n)$ when $\psi_2 = \lim_{d\to\infty}(n/d)$ is held fixed, and $\gamma > 1$. Although it is difficult to understand such trends from the mathematical definitions of $\mathscr{V}^\star$ we notice in the Figure 1 (left) that the variance decreases in overparameterized regime ($\gamma > 1$) with increasing model size ($\gamma$).

Next we study the asymptotics of $\mathscr{B}_\delta$ and $\mathscr{C}_{\beta,\delta}$ which, as discussed in lemma 3.3, quantify the part of the minority group prediction risk that comes from the model misspecification in the two group model (2.3).

**Lemma 3.8** (Misspecification error terms). *Let the assumptions 3.2, 3.4 and 3.5 hold. Following the definitions of $\mathscr{B}^\star$ and $\mathscr{V}^\star$ in lemma 3.7 we further define $\Psi_2^\star = \mathscr{B}^\star - 1 + 2(\chi + \psi)$, where $\chi$ and $\psi$ are defined in Definition 3.6. Then*

$$\lim_{d\to\infty} \mathbb{E}[\mathscr{B}_\delta] = \mathscr{M}_1^\star \triangleq \pi(1 - \pi)\mathscr{V}^\star + \pi^2\Psi_2^\star, \quad \lim_{d\to\infty} \mathbb{E}[\mathscr{C}_{\beta,\delta}] = \mathscr{M}_2^\star \triangleq \pi(\mathscr{B}^\star - 1 + \Psi_2^\star)$$

*where as in lemma 3.7, the expectation $\mathbb{E}$ is is taken over $\{x_i\}_{i=1}^n$, $\{\theta_j\}_{j=1}^N$, $\beta_0$ and $\delta$.*

Below we describe the observed trends in the bias term in the minority risk.

**Trends in the bias term**    Here, we study the asymptotic trend for the bias term $\mathscr{B}(\beta_0, \delta)$. We see from the bias decomposition in Lemma 3.3 and the term by term asymptotics in Lemmas 3.7 and 3.8 that the bias term asymptotically converges to $F_\beta^2 \mathscr{B}^\star + F_\delta^2 \mathscr{M}_1^\star + F_{\beta,\delta} \mathscr{M}_2^\star$. We mainly study the trend of this bias term in overparameterized regime with respect to the three parameters: (1) the overparametrization parameter $\gamma$, (2) the majority group proportion $\pi$ and (3) the angle between the vectors $\beta_0$ and $\beta_1$, which is denoted by $\theta$. The middle and right panels in Figure 1 give an overall understanding of the trend of the bias term with respect to the aforementioned parameters. In both of these figures we set $\|\beta_1\|_2 = \|\beta_0\|_2 = 1$. In the middle panel, we see that for a fixed angle $\theta$, the bias generally decreases with overparameterization. In contrast, for a fixed level of overparameterization $\gamma$, the bias increases as the angle $\theta$ increases. This is not surprising as intuitively, as the angle $\theta$ grows, the separation between $\beta_0$ and $\beta_1$ becomes more prominent. As a consequence, the accuracy of the model worsens and that is being reflected in the plot of the bias term.

Similar trend can also be observed when the majority group proportion $\pi$ varies for a fixed angle $\theta$ (right most panel of Figure 1). Here we set $\beta_1 = -\beta_0$, i.e., the angle $\theta = 180°$. As expected, the bias increases with parameter $\pi$ for a fixed $\gamma$ because of higher imbalance in the population. Whereas, for a fixed $\pi$, the bias decreases with increasing $\gamma$. Thus, in summary we can conclude that overparameterization generally improves the worst group performance of the model for fixed instances of $\theta$ and $\pi$.

The following theorem summarizes the asymptotic results in Lemmas 3.7 and 3.8 and states the asymptotic for minority group prediction error.

**Theorem 3.9.** *Let the assumptions 3.2, 3.4 and 3.5 hold. Following the definitions of $\mathscr{B}^\star$, $\mathscr{V}^\star$, $\mathscr{M}_1^\star$ and $\mathscr{M}_2^\star$ in Lemmas 3.7 and 3.8, we have the term by term asymptotics:*

$$\lim_{d \to \infty} \mathbb{E}[\mathscr{B}_\beta] = \mathscr{B}^\star, \ \ \lim_{d \to \infty} \mathbb{E}[\mathscr{V}(\tau)] = \tau^2 \mathscr{V}^\star, \ \ \lim_{d \to \infty} \mathbb{E}[\mathscr{B}_\delta] = \mathscr{M}_1^\star, \ and \ \lim_{d \to \infty} \mathbb{E}[\mathscr{C}_{\beta,\delta}] = \mathscr{M}_2^\star,$$

*and the minority group prediction error has the following asymptotic*

$$\mathbb{E}[R_0(\hat{a})] = F_\beta^2 \mathbb{E}[\mathscr{B}_\beta] + F_\delta^2 \mathbb{E}[\mathscr{B}_\delta] + F_{\beta,\delta} \mathbb{E}[\mathscr{C}_{\beta,\delta}] + \tau^2 \mathbb{E}[\mathscr{V}(\tau = 1)]$$
$$\to F_\beta^2 \mathscr{B}^\star + F_\delta^2 \mathscr{M}_1^\star + F_{\beta,\delta} \mathscr{M}_2^\star + \tau^2 \mathscr{V}^\star.$$

A complete proof of the theorem is provided in Appendix B.

**Trend in minority group prediction error for ERM**    To understand the trends in minority group prediction error we combine the trends in the bias and the variance terms. We again recall that we're interested in the effect of overparameterization, *i.e.* growing $\gamma$ when $\gamma > 1$. In the discussions after Lemmas 3.7 and 3.8 (and in Figure 1) we notice that both the bias and variance terms decrease or do not increase with growing overparameterization for any choices of the majority group proportion $\pi$ and the angle $\theta$ between $\beta_0$ and $\beta_1$. Since the minority risk decomposes to bias+variance terms (3.2), we notice that similar trends hold for the overall minority risk. In other words, *overparameterization improves or does not harm the minority risk*. These trends agree with the empirical findings in Pham et al. (2021).

### 3.1 LIMITING RISK FOR MAJORITY GROUP SUBSAMPLING

Though overparameterization of ML models does not harm the minority risk for ERM, they generally produce large minority risk Pham et al. (2021). To improve the risk over minority group Sagawa et al. (2020); Idrissi et al. (2021) recommend using majority group subsampling before fitting ERM and show that they achieve state-of-the-art minority risks. Here, we complement their empirical finding with a theoretical study on the minority risk for random feature models in the two group regression setup (2.3). We recall from subsection 2.3.2 that subsampling discards a randomly chosen subsample of size $n_1 - n_0$ from the majority group (of size $n_1$) to match the sample size ($n_0$) of the minority group, and then fits an ERM over the remaining sample points. Here, we repurpose the asymptotic results for ERM (Theorem 3.9) to describe the asymptotic behavior for subsampling by carefully reviewing the changes in the parameters $\psi_1$, $\psi_2$ and $\pi$. First we notice that the total sample size after subsampling is $n_S = 2n_0$ (both the majority and minority groups have sample size $n_{0,S} = n_{1,S} = n_0$) which means the new majority sample proportion is $\pi_S = n_{0,S}/n_S = 1/2$. We further notice that $\psi_1$ remains unchanged, *i.e.*, $\psi_{1,S} = \psi_1$ whereas $\psi_2 = \lim_{d \to \infty} n/d$ changes to $\psi_{2,S} =$

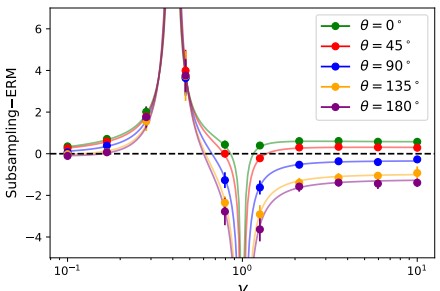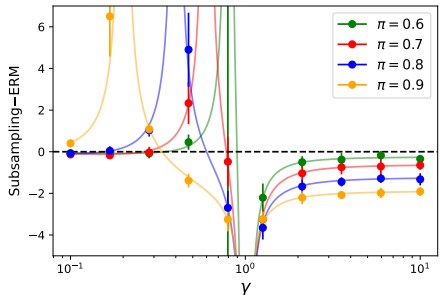

**Figure 2:** Difference in minority risk between subsampling and ERM. *Left*: the differences in minority risks for several values of $\theta$ (the angle between $\beta_0$ and $\beta_1$) and overparameterzation $\gamma$, where majority proportion $\pi$ is set at $0.8$. *Right*: the minority risk differences for several values of $\pi$ and $\gamma$ when $\theta$ is set at $180°$.

$\lim_{d\to\infty} n_S/d = \lim_{d\to\infty} 2n_0/d = \lim_{d\to\infty} 2(n_0/n) \times (n/d) = 2(1-\pi)\psi_2$. The subsampling setup is underparameterized when $n_S = 2n_0 > N$ or $\gamma_S = \psi_{1,S}/\psi_{2,S} = \gamma/(2*(1-\pi)) < 1$ and overparameterized when $\gamma_S > 1$. Rest of the setup in subsampling remains exactly the same as in the ERM setup, and the results in Lemmas 3.7, 3.8 and Theorem 3.9 continue to hold with the new parameters: $\psi_{1,S}$, $\psi_{2,S}$ and $\pi_S$.

**ERM vs subsampling**   We compare the asymptotic risks of ERM and majority group subsampling in overparameterized setting. Figure 2 shows the trend of the error difference between subsampling and ERM. Similar to the previous discussion, we denote by $\theta$ the angle between $\beta_1$ and $\beta_0$, and set $\|\beta_1\|_2 = \|\beta_0\|_2 = 1$. In the left panel, we fix the majority group parameter $\pi = 0.8$ and vary $\theta$. It is generally observed that subsampling improves the worst group performance over ERM, which is consistent with the empirical findings in Sagawa et al. (2020) and Idrissi et al. (2021). Moreover, the improvement is prominent when the group regression coefficients are well separated, i.e., $\theta$ is large. Intuitively, when $\theta$ is very large, the distinction between majority and minority groups is more prominent. As a consequence, the effect of under representation on minority group becomes more relevant, due to which the worst group performance of ERM is affected. Whereas, subsampling alleviates this effect of under representation by homogenizing the group sizes. On the other hand, the effect of under representation becomes less severe when $\beta_0$ and $\beta_1$ are close by, i.e., $\theta$ is small. In this case, we see subsampling does not improve the worst group performance significantly. In fact, for larger values of $\gamma$, subsampling performs slightly worse compared to ERM. This is again not surprising, as for smaller separation, the population structure of the two groups becomes more homogeneous. Thus, full sample ERM should deliver better performance compared to subsampling.

Similar trend is also observed in the right panel where we fix $\theta = 180°$ and vary $\pi$. Again, improvement due to subsampling is most prominent for larger values of $\pi$, which corresponds to greater imbalance in the data. Unlike the previous case, the subsampling always helps in terms of worst group performance but the improvement becomes less noticeable with decreasing $\pi$ as overparameterization grows.

## 4   RELATED WORK

**Overparameterized ML models**   The benefits of overparameterization for *average* or *overall* performance of ML models is well-studied. This phenomenon is known as "double descent", and it asserts that the (overall) risk of ML models is decreasing as model complexity increases past a certain point. This behavior has been studied empirically (Advani and Saxe, 2017; Belkin et al., 2018; Nakkiran et al., 2019; Yang et al., 2020) and theoretically (Hastie et al., 2019; Montanari et al., 2020; Mei and Montanari, 2019b; Deng et al., 2020). All these works focus on the average performance of ML models, while we focus on the performance of ML models on minority groups. The most closely related paper in this line of work is Pham et al. (2021), where they empirically study the effect of

model overparameterization on minority risk and find that the model overparamterization generally helps or does not harm minority group performance.

**Improving performance on minority groups**  There is a long line of work on improving the performance of models on minority groups. There are methods based on reweighing/subsampling (Shimodaira, 2000; Cui et al., 2019) and (group) distributionally robust optimization (DRO) (Hashimoto et al., 2018; Duchi et al., 2020; Sagawa et al., 2019). In the overparameterized regime, methods based on reweighing are not very effective (Byrd and Lipton, 2019), while subsampling has been empirically shown to improve performance on the minority groups (Sagawa et al., 2020; Idrissi et al., 2021). Our theoretical results confirm the efficacy of subsampling for improving performance on the minority groups and demonstrate that it benefits from overparameterization.

**Group fairness**  literature proposes many definition of fairness (Chouldechova and Roth, 2018), some of which require similar accuracy on the minority and majority groups (Hardt et al., 2016). Methods for achieving group fairness typically perform ERM subject to some fairness constraints (Agarwal et al., 2018). The interplay between these constraints and overparametrization is beyond the scope of this work.

## 5 SUMMARY AND DISCUSSION

In this paper, we studied the performance of overparameterized ML models on minority groups. We set up a two-group model and derived the limiting risk of random feature models in the minority group (see Theorem 3.9). In our theoretical finding we generally see that overparameterzation improves or does not harm the minority risk of ERM, which complements the findings in Pham et al. (2021). We also show theoretically that majority group subsampling is an effective way of improving the performance of overparamterized models in minority groups. This confirms the empirical results on subsampling overparameterized neural networks given in Sagawa et al. (2020) and Idrissi et al. (2021).

**What about classification?**  We observe that the trend of minority group generalization error in the random feature regression model is qualitatively distinct from its classification counterpart (see Appendix D for the model details). Figure 3 demonstrates that although overparameterization always reduces the majority error, whether or not it benefits the minority group depends on the angle between the coefficients of the two groups: overparameterzation helps the minority group generalization error if the angle is acute, while it harms the minority error if the angle is obtuse. This is different from those trends in Figure 1. The disagreement can be explained in part by the fact that the signal-to-noise ratio in classification problems is difficult to tune explicitly because the signal and noise are coupled due to the model's setup. The effect of overparametrization on minority groups under the random feature classification model is open to research by interested readers. Recent developments of convex Gaussian min-max Theorem (CGMT) and related techniques (Montanari et al., 2020; Bosch et al., 2022) could be technically useful.

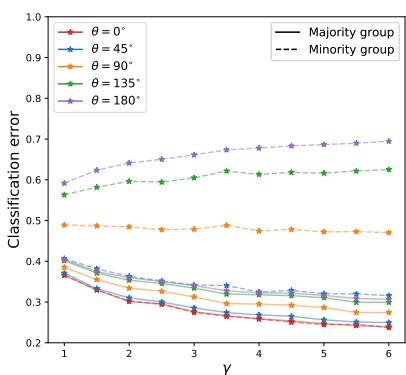

**Figure 3:** Classification error against overparamterization level $\gamma$ of the majority and minority group for various angles $\theta$ between the coefficients of two groups under the random feature classification model.

**Practical implications**  One of the main conclusion of this paper is that overparamaterization generally helps or does not harm the worst group performance of ERM. In other words, using overparametrized models is unlikely to magnify disparities across groups. However, we warn the practitioners that overparametrization *should not* be confused with a method for improving the minority group performance. Dedicated methods such as group distributionally robust optimization (Sagawa et al., 2019) and subsampling (Sagawa et al., 2020; Idrissi et al., 2021) are far more effective. In particular, subsampling improves worst-group performance and benefits from overparametrization as demonstrated in prior empirical studies (Sagawa et al., 2020; Idrissi et al., 2021) and in this paper.

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

## A  RIDGELESS REGRESSION

Similar to Section 3, here we describe analysis the linear regression model in overparameterized regime. The learner fits a linear model

$$f(x) \triangleq \beta^T x$$

to the training data, where $\beta \in \mathbf{R}^d$ is a vector of regression coefficients. We note that although the linear model is well-specified for each group, it is *misspecified* for the mixture of two groups. The learner fits a linear model to the training data in one of two ways: (1) empirical risk minimization (ERM) and (2) subsampling.

### A.1  MODEL FITTING

**Empirical risk minimization (ERM)**  The most common way of fitting a linear model is ordinary least squares (OLS):

$$\widehat{\beta}_{\text{OLS}} \in \arg\min_{\beta \in \mathbf{R}^d} \tfrac{1}{n} \sum_{i=1}^{n} \tfrac{1}{2}(y_i - \beta^T x_i)^2 = \arg\min_{\beta \in \mathbf{R}^d} \tfrac{1}{2n}\|y - X\beta\|_2^2,$$

where the rows of $X \in \mathbf{R}^{n \times d}$ are the $x_i$'s and the entries of $y \in \mathbf{R}^n$ are the $y_i$'s. We note that ERM does not use the sensitive attribute (even during training), so it is suitable for problems in which the training data does not include the sensitive attribute.

In the overparameterized case ($n < d$), $\widehat{\beta}_{\text{OLS}}$ is not well-defined because there are many linear models that can interpolate the training data. In this case, we consider the minimum $\ell_2$ norm (min-norm) linear model with zero training error:

$$\widehat{\beta}_{\min} \in \arg \min \{ \|\beta\|_2 \mid y = X\beta \} = (X^T X)^\dagger X^T y.$$

where $(X^T X)^\dagger$ denotes the Moore-Penrose pseudoinverse of $X^T X$. The min-norm least squares estimator is also known as the ridgeless least squares estimator because $\widehat{\beta}_{\min} = \lim_{\lambda \searrow 0} \widehat{\beta}_\lambda$, where

$$\widehat{\beta}_\lambda \in \arg \min_{\beta \in \mathbf{R}^d} \tfrac{1}{n} \|y - X\beta\|_2^2 + \lambda \|\beta\|_2^2.$$

We note that if $X$ has full column rank (*i.e.* $X^T X$ is non-singular), then $\widehat{\beta}_{\min}$ is equivalent to $\widehat{\beta}_{\text{OLS}}$.

**Majority group subsampling**   It is known that models trained by ERM may perform poorly on minority groups Sagawa et al. (2019). One promising way of improving performance on minority groups is majority group subsampling–randomly discarding training samples from the majority group until the two groups are equally represented in the remaining training data. This achieves an effect similar to that of upweighing the loss on minority groups in the underparameterized regime.

In the underparameterized case ($n > d$), reweighing is more statistically efficient (it does not discard training samples), so subsampling is rarely used. On the other hand, in the overparameterized case, reweighing may not have the intended effect on the performance of overparameterized models in minority groups Byrd and Lipton (2019), but Sagawa et al. (2020) show that subsampling does. Thus we also consider subsampling here as a way of improving performance in minority groups. We note that reweighing requires knowledge of the sensitive attribute, so its applicability is limited to problems in which the sensitive attribute is observed during training.

## A.2   LIMITING RISK OF ERM AND SUBSAMPLING

We are concerned with the performance of the fitted models on the minority group. In this paper, we measure performance on the minority group with the mean squared prediction error on a test sample from the minority group:

$$R_0(\beta) \triangleq \mathbf{E}\big[((\beta - \beta_0)^\top x)^2 \mid X\big] = \mathbf{E}\big[(\beta - \beta_0)^\top \Sigma (\beta - \beta_0) \mid X\big]. \tag{A.1}$$

We note that the definition of (A.1) is conditional on $X$; *i.e.* the expectation is with respect to the error terms in the training data and the features of the test sample. Although it is hard to evaluate $R_0$ exactly for finite $n$ and $d$, it is possible to approximate it with its limit in a high-dimensional asymptotic setup. In this section, we consider an asymptotic setup in which $n, d \to \infty$ in a way such that $\frac{d}{n} \to \gamma \in (0, \infty)$. If $\gamma < 1$, the problem is underparameterized; if $\gamma > 1$, it is overparameterized.

To keep things simple, we first present our results in the special case of isotropic features. We start by formally stating the assumptions on the distribution of the feature vector $P_X$.

**Assumption A.1.** *The feature vector $x \sim P_X$ has independent entries with zero mean, unit variance, and finite $8 + \epsilon$ moment for some $\epsilon > 0$.*

## A.3   EMPIRICAL RISK MINIMIZATION

We start by decomposing the minority risk (A.1) of the OLS estimator $\widehat{\beta}_{\text{OLS}}$ and the ridgeless least squares estimator $\widehat{\beta}_{\min}$. Let $n_0$ and $n_1$ be the number of training examples from the minority and the majority groups respectively. Without loss of generality, we arrange the training samples so that the first $n_0$ examples are those from the minority group:

$$X = \begin{bmatrix} X_0 \\ X_1 \end{bmatrix}, \quad y = \begin{bmatrix} y_0 \\ y_1 \end{bmatrix}.$$

**Lemma A.2** (ERM minority risk decomposition). *Let $\widehat{\beta}$ be either $\widehat{\beta}_{OLS}$ or $\widehat{\beta}_{\min}$. We have*

$$
\begin{aligned}
R_0(\widehat{\beta}) &= \mathbf{E}\big[(\widehat{\beta} - \beta_0)^\top(\widehat{\beta} - \beta_0)\big] \\
&= \beta_0^\top(I_d - \Pi_X)\beta_0 + \frac{\tau^2}{n}Tr(\widehat{\Sigma}^\dagger) + \delta^\top\Big\{\begin{bmatrix}0_{n_0 \times d} \\ X_1\end{bmatrix}^\top X/n\Big\}(\widehat{\Sigma}^\dagger)^2\Big\{X^\top\begin{bmatrix}0_{n_0 \times d} \\ X_1\end{bmatrix}/n\Big\}\delta \\
&\quad + 2\delta^\top\Big\{\begin{bmatrix}0_{n_0 \times d} \\ X_1\end{bmatrix}^\top X/n\Big\}(\widehat{\Sigma}^\dagger)^2\widehat{\Sigma}\beta_0
\end{aligned}
\tag{A.2}
$$

*where $\Pi_X \triangleq X^\dagger X$ is the projector onto $\mathsf{ran}(X^\top)$, $\delta \triangleq \beta_0 - \beta_1$, and $\widehat{\Sigma} \triangleq \frac{1}{n}X^\top X$ is the sample covariance matrix of the features in the training data.*

**Inductive bias**   We recognize the first term on the right side of (A.2) as a squared bias term. This term reflects the inductive bias of ridgeless least squares: it is orthogonal to $\ker(X)$, so it cannot capture the part of $\beta_0$ in $\ker(X)$. We note that this term is only non-zero in the overparameterized regime: $\widehat{\beta}_{\mathrm{OLS}}$ has no inductive bias.

**Lemma A.3** (Hastie et al. (2019), Lemma 2). *In addition to Assumption A.1, assume $\|\beta_0\|_2^2 = s_0$ for all $n, d$. We have*

$$
\beta_0^\top(I_d - \Pi_X)\beta_0 \xrightarrow{p} \big\{s_0\big(1 - \tfrac{1}{\gamma}\big)\big\} \vee 0 \text{ as } n, d \to \infty, \tfrac{d}{n} \to \gamma.
$$

**Variance**   The second term on the right side of (A.2) is a variance term. The limit of this term in the high-dimensional asymptotic setting is known.

**Lemma A.4** (Hastie et al. (2019), Theorem 1, Lemma 3). *Under Assumption A.1, we have*

$$
\frac{\tau^2}{n}Tr(\widehat{\Sigma}^\dagger) \xrightarrow{p} \begin{cases} \tau^2\frac{\gamma}{1-\gamma} & \gamma < 1 \\ \frac{\tau^2}{\gamma-1} & \gamma > 1 \end{cases} \text{ as } n, d \to \infty, \tfrac{d}{n} \to \gamma.
$$

**Approximation error**   The third and forth terms in (A.2) reflects the approximation error of $\widehat{\beta}_{\mathrm{OLS}}$ and $\widehat{\beta}_{\min}$ because the linear model is misspecified for the mixture of two groups. Unlike the inductive bias and variance terms, this term does not appear in prior studies of the average/overall risk of the ridgeless least squares estimator Hastie et al. (2019).

**Lemma A.5.** *In addition to Assumption A.1, assume $\|\delta\|_2^2 = r$ and $\delta^\top\beta_0 = c$ for all $n, d$. As $d \to \infty$ we have*

$$
\delta^\top\Big\{\begin{bmatrix}0_{n_0 \times d} \\ X_1\end{bmatrix}^\top X/n\Big\}(\widehat{\Sigma}^\dagger)^2\Big\{X^\top\begin{bmatrix}0_{n_0 \times d} \\ X_1\end{bmatrix}/n\Big\}\delta \xrightarrow{p} \begin{cases} r\frac{\pi\gamma}{1-\gamma} + r\frac{\pi^2(1-2\gamma)}{1-\gamma} & \gamma < 1 \\ r\frac{\pi}{\gamma-1} + r\frac{\pi^2(\gamma-2)}{\gamma(\gamma-1)} & \gamma > 1 \end{cases}
$$

*and*

$$
\delta^\top\Big\{\begin{bmatrix}0_{n_0 \times d} \\ X_1\end{bmatrix}^\top X/n\Big\}(\widehat{\Sigma}^\dagger)^2\widehat{\Sigma}\beta_0 \xrightarrow{p} c\pi(\{1/\gamma\} \wedge 1).
$$

Before moving on, we note that (the limit of) the approximation error term is the only term that depends on the majority fraction $\pi$. Though the inductive bias may increase at overparameterized regime ($\gamma > 1$) with growing $\gamma$ when the SNR ($= \|\beta_0\|_2^2/\tau^2$) is large, we notice that the approximation error terms always decrease with growing overparameterization. In fact they tend to zero as $\gamma \to \infty$.

We plot the (the limit of) the minority risk prediction error (MSPE) in Figure 4. In both overparameterized and underparameterized regimes, the MSPE increases as $\pi$ increases. This is expected: as the fraction of training samples from the minority group decreases, we expect ERM to train a model that aligns more closely with the regression function of the majority group.

We also notice that in the overparameterized regime ($\gamma > 1$) when the SNR ($\|\beta_2\|_2^2/\tau^2$) is high the MSPE increases with growing $\gamma$ if the two groups are aligned (the angle between $\beta_0$ and $\beta_1$ is $\theta = 0$). This trend is also observed in Hastie et al. (2019).

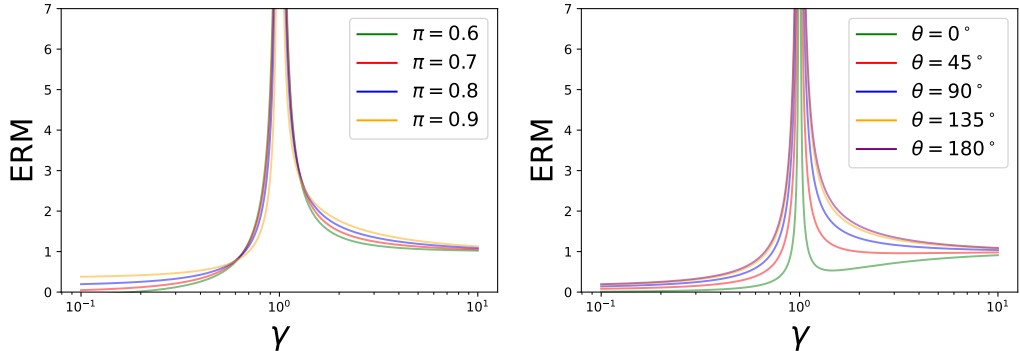

**Figure 4:** Minority group prediction error for the ridgeless least square ERM in the isotropic features case. The *left plot* considers the setup with varying $\pi$ when the angle between $\beta_0$ and $\beta_1$ is set at $\theta = 180°$. The right plot sets $\pi = 0.8$ and considers different values of $\theta$. Here the SNR is set at $\|\beta_0\|_2^2/\tau^2 = \|\beta_1\|_2^2/\tau^2 = 10$.

### A.4 INADEQUACIES OF RIDGELESS LEAST SQUARES

There is one notable disagreement between the asymptotic risk of the ridgeless least squares estimator and the empirical practice Pham et al. (2021) in modern ML: the risk of the ridgeless least squares estimator *increases* as the overparameterization ratio $\gamma$ increases, while the accuracy of modern ML models generally improves with overparameterization. This has led ML practitioner to train overparameterized models whose risks exhibit a double-descent phenomenon Belkin et al. (2018). Inspecting the asymptotic risk of ridgeless least squares reveals the increase in risk (as $\gamma$ increases) is due to the inductive bias term (see A.3). For high SNR problems ($s_0 \gg \tau^2$), the increase of the inductive bias term dominates the decrease of the variance term (see A.4), which leads the overall risk to increase. This is a consequence of the fact that problem dimension (the dimension of the inputs) and the degree of overparameterization are tied ridgeless least squares. In order to elucidate behavior (in the risk) that more closely matches empirical observations, we study the random features models, which allows us to keep the problem dimension fixed while increasing the overparameterization by increasing the number of random features.

### A.5 MAJORITY GROUP SUBSAMPLING

We note that the (limiting) minority risk curve of majority group subsampling is the (limiting) minority risk curve of ERM after a change of variables. Indeed, it is not hard to check that discarding training sample from the majority group until the groups are balanced in the training data leads to a reduction in total sample size by a factor of $2(1 - \pi)$ (recall $\pi \in [\frac{1}{2}, 1]$). In other words, if the (group imbalanced) training data consists of $n$ samples, then the (group balanced) training data will have $2(1 - \pi)$ training samples. In the $n, d \to \infty$, $\frac{d}{n} \to \gamma$ limit, this is equivalent to increasing $\gamma$ by a factor of $\frac{1}{2(1-\pi)}$. The limiting results in ERM can be reused with the following changes: under subsampling (1) $\pi$ changes to $1/2$, (2) $\gamma$ changes to $\gamma/(2 - 2\pi)$, and (3) rest of the parameters for the limit remains same.

In Figure 5, where we plot the differences between subsampling and ERM minority risk, we observe that subsampling generally improves minority group performance over ERM. This aligns with the empirical findings in Sagawa et al. (2020); Idrissi et al. (2021). The only instance when subsampling has worse minority risk than ERM is when the angle between $\beta_0$ and $\beta_1$ is zero, *i.e.* $\beta_0 = \beta_1$. This perfectly agrees with intuition; when there is no differnce between the distributions of the two groups then subsampling discards some samples for majority group which are valuable in learning the predictor for minority group, resulting an inferior predictor for minority group.

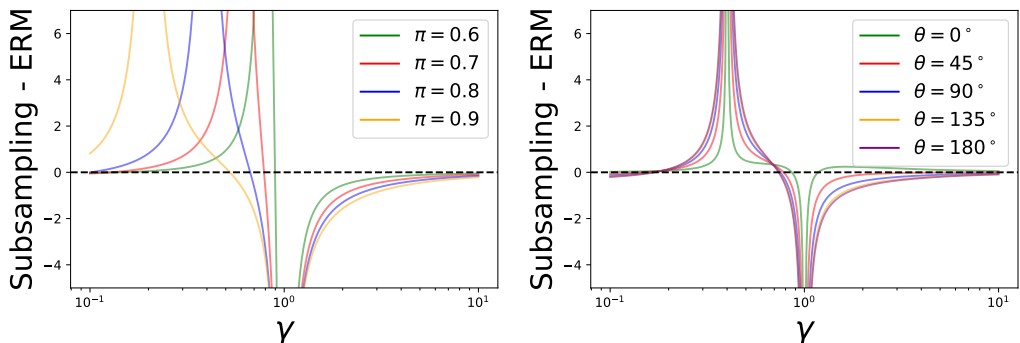

**Figure 5:** Minority error differnce between subsampling and ERM for the ridgeless least square in the isotropic features case. The *left plot* considers the setup with varying $\pi$ when the angle between $\beta_0$ and $\beta_1$ is set at $\theta = 180°$. The right plot sets $\pi$ at 0.8 and consider different values of $\theta$. Here the SNR is set at $\|\beta_0\|_2^2/\tau^2 = \|\beta_1\|_2^2/\tau^2 = 5$.

### A.6 PROOFS OF LEMMAS A.2 AND A.5

#### A.6.1 PROOF OF LEMMA A.2

Denoting $\mathbf{U} = (X^\top X/n)^\dagger X/n$ we note that the estimation error $\widehat{\beta} - \beta_0$ is

$$\widehat{\beta} - \beta_0 = \mathbf{U}(X\beta_0 + \begin{bmatrix} 0_{n_0 \times d} \\ X_1 \end{bmatrix}\delta + \epsilon) - \beta_0 = (\Pi_X - I_d)\beta_0 + \mathbf{U}\begin{bmatrix} 0_{n_0 \times d} \\ X_1 \end{bmatrix}\delta + \mathbf{U}\epsilon,$$

we have the following decomposition of ERM minority risk

$$
\begin{aligned}
&R_0(\widehat{\beta}) \\
=&\mathbf{E}\big[(\widehat{\beta} - \beta_0)^\top(\widehat{\beta} - \beta_0)\big] \\
=&\mathbf{E}\left[\left\{(\Pi_X - I_d)\beta_0 + \mathbf{U}\begin{bmatrix} 0_{n_0 \times d} \\ X_1 \end{bmatrix}\delta + \mathbf{U}\epsilon\right\}^\top\left\{(\Pi_X - I_d)\beta_0 + \mathbf{U}\begin{bmatrix} 0_{n_0 \times d} \\ X_1 \end{bmatrix}\delta + \mathbf{U}\epsilon\right\}\right] \\
=&\underbrace{\beta_0^\top(\Pi_X - I_d)^2\beta_0}_{\text{I}} + \underbrace{\mathbf{E}\big[\epsilon^\top(\mathbf{U})^\top\mathbf{U}\epsilon\big]}_{\text{II}} + \underbrace{\delta^\top\begin{bmatrix} 0_{n_0 \times d} \\ X_1 \end{bmatrix}^\top(\mathbf{U})^\top\mathbf{U}\begin{bmatrix} 0_{n_0 \times d} \\ X_1 \end{bmatrix}\delta}_{\text{III}} + \\
&\underbrace{2\delta^\top\begin{bmatrix} 0_{n_0 \times d} \\ X_1 \end{bmatrix}^\top(\mathbf{U})^\top(\Pi_X - I_d)\beta_0}_{\text{IV}} + \underbrace{2\mathbf{E}\left[\left\{(\Pi_X - I_d)\beta_0 + \mathbf{U}\begin{bmatrix} 0_{n_0 \times d} \\ X_1 \end{bmatrix}\delta\right\}^\top\mathbf{U}\epsilon\right]}_{\text{V}}.
\end{aligned}
$$

The term I is equal to $\beta_0^\top(\Pi_X - I_d)\beta_0$ because $\Pi_X - I_d$ is a projection matrix. By the linearity of expectation and trace operator, the term II is equal to

$$\mathbf{E}\big[\text{Tr}(\epsilon^\top(\mathbf{U})^\top\mathbf{U}\epsilon)\big] = \text{Tr}\left\{\frac{1}{n}\widehat{\Sigma}^\dagger\mathbf{E}[\epsilon\epsilon^\top]\right\} = \text{Tr}\left\{\frac{1}{n}\widehat{\Sigma}^\dagger\tau^2 I_n\right\} = \frac{\tau^2}{n}\text{Tr}(\widehat{\Sigma}^\dagger).$$

By properties of Moore–Penrose pseudoinverse, the term III and IV are equal to

$$\delta^\top\left\{\begin{bmatrix} 0_{n_0 \times d} \\ X_1 \end{bmatrix}^\top X/n\right\}(\widehat{\Sigma}^\dagger)^2\left\{X^\top\begin{bmatrix} 0_{n_0 \times d} \\ X_1 \end{bmatrix}/n\right\}\delta \quad \text{and} \quad 2\delta^\top\left\{\begin{bmatrix} 0_{n_0 \times d} \\ X_1 \end{bmatrix}^\top X/n\right\}(\widehat{\Sigma}^\dagger)^2\widehat{\Sigma}\beta_0$$

respectively. The term V is 0 since $\mathbf{E}[\epsilon] = 0$. Hence we complete the proof. $\qquad\square$

#### A.6.2 PROOF OF LEMMA A.5

We first prove for the cross term that

$$\mathbf{E}[\beta_0^\top\widehat{\Sigma}(\widehat{\Sigma}^\dagger)^2\frac{1}{n}X^\top(0, X_1^\top)^\top\delta] = \frac{\beta_0^\top\delta}{d}\mathbf{E}[\text{Tr}\{\widehat{\Sigma}(\widehat{\Sigma}^\dagger)^2\frac{1}{n}X^\top(0, X_1^\top)^\top\}]. \tag{A.3}$$

To realize the above we notice that for any orthogonal matrix $O$ it holds

$$\mathbf{E}[\beta_0^\top \widehat{\Sigma}(\widehat{\Sigma}^\dagger)^2 \frac{1}{n} X^\top (0, X_1^\top)^\top \delta] = \mathbf{E}[\beta_0^\top O \widehat{\Sigma}_O (\widehat{\Sigma}_O^\dagger)^2 \frac{1}{n} X_O^\top (0, X_{O,1}^\top)^\top O^\top \delta]$$

where $X$ is replaced by $X_O = XO$ and the replacement in $X$ changes the covariance matrix to $\widehat{\Sigma}_O$ and it's Moore-Penrose inverse to $\widehat{\Sigma}_O^\dagger$. Noticing that $X \stackrel{d}{=} X_O$, $\widehat{\Sigma} \stackrel{d}{=} \widehat{\Sigma}_O$ and $\widehat{\Sigma}^\dagger \stackrel{d}{=} \widehat{\Sigma}_O^\dagger$ we conclude

$$\mathbf{E}[\beta_0^\top O \widehat{\Sigma}_O (\widehat{\Sigma}_O^\dagger)^2 \frac{1}{n} X_O^\top (0, X_{O,1}^\top)^\top O^\top \delta] = \mathbf{E}[\beta_0^\top O \widehat{\Sigma}(\widehat{\Sigma}^\dagger)^2 \frac{1}{n} X^\top (0, X_1^\top)^\top O^\top \delta].$$

We now let $O$ to be uniformly distributed over the set of all $d \times d$ orthogonal matrices (distributed according to Haar measure) which results in

$$\mathbf{E}[\beta_0^\top O \widehat{\Sigma}(\widehat{\Sigma}^\dagger)^2 \frac{1}{n} X^\top (0, X_1^\top)^\top O^\top \delta] = \mathbf{E}[\mathrm{Tr}\{\ \widehat{\Sigma}(\widehat{\Sigma}^\dagger)^2 \frac{1}{n} X^\top (0, X_1^\top)^\top O^\top \delta \beta_0^\top O\}]$$

$$= \mathbf{E}[\mathrm{Tr}\{\ \widehat{\Sigma}(\widehat{\Sigma}^\dagger)^2 \frac{1}{n} X^\top (0, X_1^\top)^\top (\delta^\top \beta_0 / d)\}].$$

To calculate the term

$$\frac{1}{d}\mathbf{E}[\mathrm{Tr}\{\ \widehat{\Sigma}(\widehat{\Sigma}^\dagger)^2 \frac{1}{n} X^\top (0, X_1^\top)^\top\}] = \frac{1}{d}\mathbf{E}[\mathrm{Tr}\{\ \widehat{\Sigma}(\widehat{\Sigma}^\dagger)^2 (\frac{1}{n} X_1^\top X_1)\}]$$

we notice that

$$\frac{1}{d}\mathbf{E}[\mathrm{Tr}\{\ \widehat{\Sigma}(\widehat{\Sigma}^\dagger)^2 (\frac{1}{n} X_1^\top X_1)\}] = \frac{1}{d}\sum_{i=n_0+1}^{n} \mathbf{E}[\mathrm{Tr}\{\ \widehat{\Sigma}(\widehat{\Sigma}^\dagger)^2 (\frac{1}{n} x_i x_i^\top)\}]$$

and each of the terms $\mathrm{Tr}\{\ \widehat{\Sigma}(\widehat{\Sigma}^\dagger)^2 (\frac{1}{n} x_i x_i^\top)\}$ are identically distributed. We appeal to exchangability argument to conclude

$$\frac{1}{d}\sum_{i=n_0+1}^{n} \mathbf{E}[\mathrm{Tr}\{\ \widehat{\Sigma}(\widehat{\Sigma}^\dagger)^2 (\frac{1}{n} x_i x_i^\top)\}] = \pi \frac{1}{d}\sum_{i=1}^{n} \mathbf{E}[\mathrm{Tr}\{\ \widehat{\Sigma}(\widehat{\Sigma}^\dagger)^2 (\frac{1}{n} x_i x_i^\top)\}]$$

$$= \pi \frac{1}{d}\mathbf{E}[\mathrm{Tr}\{\ \widehat{\Sigma}(\widehat{\Sigma}^\dagger)^2 \widehat{\Sigma}\}]$$

where we notice that $\widehat{\Sigma}\widehat{\Sigma}^\dagger$ is a projection matrix and conclude

$$\pi \frac{1}{d}\mathbf{E}[\mathrm{Tr}\{\ \widehat{\Sigma}(\widehat{\Sigma}^\dagger)^2 \widehat{\Sigma}\}] = \pi \frac{1}{d}\mathbf{E}[\mathrm{Tr}\{\ \widehat{\Sigma}\widehat{\Sigma}^\dagger\}].$$

If $d < n$ *i.e.* $\gamma < 1$ then $\widehat{\Sigma}$ is invertible and and it holds $\widehat{\Sigma}\widehat{\Sigma}^\dagger = \mathbb{I}_d$. If $d > n$ then there are exactly $n$ many non-zero eigen-values in $\widehat{\Sigma}\widehat{\Sigma}^\dagger$ and it holds $\mathrm{Tr}\{\ \widehat{\Sigma}\widehat{\Sigma}^\dagger\} = n$. Combining the results we obtain

$$\frac{1}{d}\mathbf{E}[\mathrm{Tr}\{\ \widehat{\Sigma}\widehat{\Sigma}^\dagger\}] = \{1/\gamma\} \wedge 1.$$

Now, to calculate the term

$$\mathbf{E}[\delta^\top \{(0, X_1^\top) X/n\}(\widehat{\Sigma}^\dagger)^2 \{X^\top (0, X_1^\top)^\top \delta]$$

we first use similar argument to (A.3) and obtain

$$\mathbf{E}[\delta^\top \{(0, X_1^\top) X/n\}(\widehat{\Sigma}^\dagger)^2 \{X^\top (0, X_1^\top)^\top \delta] = \|\delta\|_2^2 \frac{1}{d}\mathbf{E}[\mathrm{Tr}\{\frac{X_1^\top X_1}{n}(\widehat{\Sigma}^\dagger)^2 \frac{X_1^\top X_1}{n}\}]$$

We rewrite

$$\frac{1}{d}\mathbf{E}[\mathrm{Tr}\{\frac{X_1^\top X_1}{n}(\widehat{\Sigma}^\dagger)^2 \frac{X_1^\top X_1}{n}\}] = n_1 F_{1,1} + n_1(n_1 - 1) F_{1,2}$$

where we define

$$F_{1,1} \triangleq \frac{1}{d}\mathbf{E}[\mathrm{Tr}\{\frac{x_1 x_1^\top}{n}(\widehat{\Sigma}^\dagger)^2 \frac{x_1 x_1^\top}{n}\}]$$

$$F_{1,2} \triangleq \frac{1}{d}\mathbf{E}[\mathrm{Tr}\{\frac{x_1 x_1^\top}{n}(\widehat{\Sigma}^\dagger)^2 \frac{x_2 x_2^\top}{n}\}].$$

To calculate $nF_{1,1}$ we notice that

$$nF_{1,1} = \frac{n}{d}\mathbf{E}[\mathrm{Tr}\{\frac{x_1 x_1^\top}{n}(\widehat{\Sigma}^\dagger)^2\frac{x_1 x_1^\top}{n}\}]$$

$$= \frac{1}{\gamma}\frac{x_1^\top x_1}{n}\frac{x_1^\top(\widehat{\Sigma}^\dagger)^2 x_1}{n}.$$

Denoting $\xrightarrow{L_1}$ as the $L_1$ convergence we notice that $\frac{x_1 x_1^\top}{n} \xrightarrow{L_1} \gamma$, *i.e.* $\mathbf{E}[(\|x_1\|_2^2/n - \gamma)^2] \to 0$. We now calculate the convergence limit for $\frac{x_1^\top(\widehat{\Sigma}^\dagger)^2 x_1}{n}$. Noticing that

$$\frac{1}{n}x_1^\top(\widehat{\Sigma}^\dagger)^2 x_1 = \lim_{z\to 0+}\frac{1}{n}x_1^\top(\widehat{\Sigma} + z\mathbb{I}_d)^{-2}x_1$$

we write $\widehat{\Sigma} + z\mathbb{I}_d = (x_1 x_1^\top/n) + A_z$, where $A_z = X_{-1}^\top X_{-1}/n + z\mathbb{I}_d$. Using the Woodbery decomposition

$$\{(x_1 x_1^\top/n) + A_z\}^{-1} = A_z^{-1} - \frac{A_z^{-1}(x_1 x_1^\top/n)A_z^{-1}}{1 + x_1^\top A_z^{-1}x_1}$$

we obtain that

$$\frac{1}{n}x_1^\top(\widehat{\Sigma} + z\mathbb{I}_d)^{-2}x_1 = \frac{\frac{1}{n}x_1^\top A_z^{-2}x_1}{(1 + \frac{1}{n}x_1^\top A_z^{-1}x_1)^2}$$

In Lemma A.6 we notice that

$$\mathbf{E}[\frac{1}{n}x_1^\top A_z^{-1}x_1] \to \begin{cases} \frac{\gamma}{1-\gamma} & \gamma < 1 \\ \frac{1}{\gamma-1} & \gamma > 1 \end{cases}, \quad \mathbf{E}[\frac{1}{n}x_1^\top A_z^{-2}x_1] \to \begin{cases} \frac{\gamma}{(1-\gamma)^3} & \gamma < 1 \\ \frac{\gamma}{(\gamma-1)^3} & \gamma > 1 \end{cases} \quad \text{as } d \to \infty.$$

$$\mathrm{var}[\frac{1}{n}x_1^\top A_z^{-1}x_1], \mathrm{var}[\frac{1}{n}x_1^\top A_z^{-2}x_1] \to 0$$

which implies $nF_{1,1}$ converges to

$$nF_{1,1} \xrightarrow{L_1} \begin{cases} \frac{\gamma}{1-\gamma} & \gamma < 1 \\ \frac{1}{\gamma(\gamma-1)} & \gamma > 1 \end{cases}$$

Noticing that

$$\frac{1}{d}\mathbf{E}[\mathrm{Tr}\{\widehat{\Sigma}\widehat{\Sigma}^\dagger\}] = \frac{1}{d}\mathbf{E}[\mathrm{Tr}\{\widehat{\Sigma}(\widehat{\Sigma}^\dagger)^2\widehat{\Sigma}\}] = nF_{1,1} + n(n-1)F_{1,2}$$

we obtain the convergence of $n(n-1)F_{1,2}$ as

$$n(n-1)F_{1,2} \xrightarrow{L_1} \begin{cases} \frac{1-2\gamma}{1-\gamma} & \gamma < 1 \\ \frac{\gamma-2}{\gamma(\gamma-1)} & \gamma > 1 \end{cases}$$

which finally yields

$$\frac{1}{d}\mathbf{E}[\mathrm{Tr}\{\frac{X_1^\top X_1}{n}(\widehat{\Sigma}^\dagger)^2\frac{X_1^\top X_1}{n}\}] = n_1 F_{1,1} + n_1(n_1-1)F_{1,2}$$

$$\asymp \pi n F_{1,1} + \pi^2 n(n-1)F_{1,2}$$

$$\xrightarrow{L_1} \begin{cases} \frac{\pi\gamma}{1-\gamma} + \frac{\pi^2(1-2\gamma)}{1-\gamma} & \gamma < 1 \\ \frac{\pi}{\gamma-1} + \frac{\pi^2(\gamma-2)}{\gamma(\gamma-1)} & \gamma > 1 \end{cases}$$

**Lemma A.6.** *As $z \to 0+$ and $d \to \infty$ we have*

$$\mathbf{E}[\frac{1}{n}x_1^\top A_z^{-1}x_1] \to \begin{cases} \frac{\gamma}{1-\gamma} & \gamma < 1 \\ \frac{1}{\gamma-1} & \gamma > 1 \end{cases},$$

$$\mathbf{E}[\frac{1}{n}x_1^\top A_z^{-2}x_1] \to \begin{cases} \frac{\gamma}{(1-\gamma)^3} & \gamma < 1 \\ \frac{\gamma}{(\gamma-1)^3} & \gamma > 1 \end{cases}$$

*and*

$$var[\frac{1}{n}x_1^\top A_z^{-1}x_1], var[\frac{1}{n}x_1^\top A_z^{-2}x_1] \to 0$$

*Proof of Lemma A.6.* To prove the results about variances we first establish that for any symmetric matrix $B = ((b_{ij})) \in \mathbf{R}^{d \times d}$ it holds

$$\text{var}(x_1^\top B x_1) \leq c \text{Tr}\{B^2\}, \tag{A.4}$$

for some $c > 0$. Writing $x_1 = (u_1, \ldots, u_d)^\top$ we see that

$$\text{var}(x_1^\top B x_1) = \text{var}\big(\textstyle\sum_{i,j} u_i u_j b_{ij}\big)$$
$$= \sum_{i,j,k,l} b_{ij} b_{kl} \text{cov}(u_i u_j, u_k u_l)$$

Noticing that

$$\text{cov}(u_i u_j, u_k u_l) = \mathbf{E}[u_i u_j u_k u_l] - \mathbf{E}[u_i u_j]\mathbf{E}[u_k u_l]$$

we see that

1. If one of $i, j, k, l$ is distinct from the others then $\text{cov}(u_i u_j, u_k u_l) = 0$.

2. If $i = j$ and $k = l$ and $i \neq k$ then

$$\mathbf{E}[u_i u_j u_k u_l] - \mathbf{E}[u_i u_j]\mathbf{E}[u_k u_l] = \mathbf{E}[u_i^2 u_k^2] - \mathbf{E}[u_i^2]\mathbf{E}[u_k^2] = 0.$$

3. If $i = j = k = l$ then $\text{cov}(u_i u_j, u_k u_l) = \text{var}(u_1^2)$.

4. If $\{i = k$ and $j = l$ and $i \neq j\}$ or $\{i = l$ and $j = k$ and $i \neq j\}$ then we have

$$\mathbf{E}[u_i u_j u_k u_l] - \mathbf{E}[u_i u_j]\mathbf{E}[u_k u_l] = \text{var}(u_i u_j) = \text{var}(u_1 u_2).$$

Gathering the terms we notice that

$$\text{var}(x_1^\top B x_1) = \sum_i \text{var}(u_i^2) b_{ii}^2 + 2 \sum_{i \neq j} b_{ij}^2 \text{var}(u_i u_j)$$
$$\leq c \sum_{i,j} b_{ij}^2 = c \text{Tr}\{B^2\},$$

where $c = 2\{\text{var}(u_1^2) \vee \text{var}(u_1 u_2)\}$.

Using (A.4) we notice that

$$\lim_{z \to 0+} \text{var}(x_1^\top A_z^{-j} x_1 / n) = \frac{1}{n^2} \text{Tr}\{(\widehat{\Sigma}_{-1}^\dagger)^{2j}\} \asymp \frac{1}{n^2} \text{Tr}\{(\widehat{\Sigma}^\dagger)^{2j}\} \to 0.$$

We notice that

$$\mathbf{E}[\frac{1}{n} x_1^\top A_z^{-1} x_1] = \frac{1}{n} \text{Tr}\{A_z^{-1}\} \asymp \frac{1}{n} \text{Tr}\{\widehat{\Sigma}^\dagger\}$$

which appears in the variance term in Lemma A.4, and we conclude that

$$\mathbf{E}[\frac{1}{n} x_1^\top A_z^{-1} x_1] \to \begin{cases} \frac{\gamma}{1-\gamma} & \gamma < 1 \\ \frac{1}{\gamma - 1} & \gamma > 1 \end{cases} \text{ as } d \to \infty.$$

To calculate

$$\mathbf{E}[\frac{1}{n} x_1^\top A_z^{-2} x_1] = \frac{1}{n} \text{Tr}\{A_z^{-2}\} \asymp \frac{1}{n} \text{Tr}\{(\widehat{\Sigma}^\dagger)^2\}$$

we consider case by case.

For $\gamma < 1$ we see that

$$\frac{1}{n} \text{Tr}\{(\widehat{\Sigma}^\dagger)^2\} = \gamma \frac{1}{d} \text{Tr}\{(\widehat{\Sigma})^{-2}\}$$
$$\to \gamma \int \frac{1}{s^2} dF_\gamma(s)$$

The Stieltjes transformation of $\mu_\gamma$ is

$$
\begin{aligned}
s_\gamma(z) &= \int \frac{1}{x-z} d\mu_\gamma(x) \\
&= \frac{1-\gamma-z-\sqrt{(1-\gamma-z)^2-4\gamma z}}{2\gamma z}, \quad z \in \mathbb{C}\backslash[1-\sqrt{\gamma}, 1+\sqrt{\gamma}].
\end{aligned}
$$

For $|z| < 1-\sqrt{\gamma}$

$$
\begin{aligned}
s_\gamma(z) &= \int \frac{1}{x-z} d\mu_\gamma(x) \\
&= \int \frac{1}{x} \frac{1}{1-z/x} d\mu_\gamma(x) \\
&= \int \frac{1}{x} \sum_{k=0}^{\infty} (z/x)^k d\mu_\gamma(x)
\end{aligned}
$$

Hence, we have

$$
\lim_{z \to 0-} \frac{d}{dz}\big(s_\gamma(z)\big) = \mathbf{E}[1/X^2].
$$

Now,

$$
\begin{aligned}
&1-\gamma-z-\sqrt{(1-\gamma-z)^2-4\gamma z} \\
&= 1-\gamma-z-\big(1+\gamma^2+z^2-2\gamma-2z+2\gamma z-4\gamma z\big)^{1/2} \\
&= 1-\gamma-z-\big((1-\gamma)^2+z^2-2z-2\gamma z\big)^{1/2} \\
&\asymp 1-\gamma-z-(1-\gamma)\Big[1+\frac{z^2-2z-2\gamma z}{2(1-\gamma)^2} - \frac{(z^2-2z-2\gamma z)^2}{8(1-\gamma)^4}\Big], \quad \text{for } z \to 0- \\
&\asymp 1-\gamma-z-(1-\gamma)\Big[1+\frac{z^2-2z-2\gamma z}{2(1-\gamma)^2} - \frac{z^2(1+\gamma)^2}{2(1-\gamma)^4}\Big], \quad \text{for } z \to 0- \\
&= \frac{z^2(1+\gamma)^2}{2(1-\gamma)^3} - \frac{z^2-4\gamma z}{2(1-\gamma)} \\
&= \frac{2z^2\gamma}{(1-\gamma)^3} + \frac{2\gamma z}{1-\gamma}
\end{aligned}
$$

which implies

$$
s_\gamma(z) \asymp \frac{z}{(1-\gamma)^3} + \frac{1}{1-\gamma}
$$

This establish that

$$
\frac{1}{n}\mathrm{Tr}\{(\widehat{\Sigma}^\dagger)^2\} = \gamma \lim_{z \to 0-} \frac{d}{dz}\big(s_\gamma(z)\big) = \frac{\gamma}{(1-\gamma)^3}\,.
$$

For $\gamma > 1$ we notice that

$$
\frac{1}{n}\mathrm{Tr}\{(\widehat{\Sigma}^\dagger)^2\} = \frac{1}{n}\sum_{i=1}^{n} \frac{1}{s_i^2}
$$

where $s_i$'s are non-zero eigen-values of $X^\top X/n$. This is also equal to

$$
\frac{1}{\gamma p}\sum_{i=1}^{n} \frac{1}{t_i^2}
$$

where $t_i$ are the eigen-values of the invertible matrix $XX^\top/d$. Hence we have

$$
\frac{1}{n}\mathrm{Tr}\{(\widehat{\Sigma}^\dagger)^2\} = \frac{1}{\gamma^2}\frac{1}{n}\mathrm{Tr}\{(XX^\top/d)^{-2}\} \to \frac{1}{\gamma^2} \lim_{z \to 0+} \frac{d}{dz}\big(s_{1/\gamma}(z)\big) = \frac{\gamma}{(\gamma-1)^3}\,.
$$

We combine the limits for $\gamma < 1$ and $\gamma > 1$ to write to write

$$\mathbf{E}[\frac{1}{n}x_1^\top A_z^{-2} x_1] \to \begin{cases} \frac{\gamma}{(1-\gamma)^3} & \gamma < 1 \\ \frac{\gamma}{(\gamma-1)^3} & \gamma > 1 \end{cases} \text{ as } d \to \infty.$$

$\square$

## B  PROOF OF THEOREM 3.9

Before diving into the main proof we give a rough sketch of the whole proof. Essentially, the proof has three major components.

1. In the first part, we establish the limiting risk where we treat $\beta_0$ and $\delta$ as uncorrelated random variables. (Section B.1)

2. In the second part of the proof we establish the necessary bias-variance decomposition under orthogonality of the fixed parameters $\beta_0$ and $\delta$. (Section B.2)

3. In the final part, we use the results of previous two parts to establish the limiting risk result for $\beta_0$ and $\delta$ in general position. (Section B.3 and B.4)

### B.1  LIMITING RISK FOR UNCORRELATED PARAMETERS

We start this section with some definitions of some important quantities.

**Definition B.1.** *Let the functions $\nu_1, \nu_2 : \mathbb{C}_+ \to \mathbb{C}_+$ be uniquely defined by the following conditions: (i) $\nu_1, \nu_2$ are analytic on $\mathbb{C}_+$. (ii) For $Im(\zeta) > 0$, $\nu_1(\zeta)$ and $\nu_2(\zeta)$ satisfy the equations*

$$\nu_1 = \psi_1 \left( -\zeta - \nu_2 - \frac{\xi^2 \nu_2}{1 - \xi^2 \nu_1 \nu_2} \right)^{-1}$$

$$\nu_2 = \psi_1 \left( -\zeta - \nu_1 - \frac{\xi^2 \nu_1}{1 - \xi^2 \nu_1 \nu_2} \right)^{-1}$$

*(iii) $(\nu_1(\zeta), \nu_2(\zeta))$ is the unique solution of these equations with*

$$|\nu_1(\zeta)| \le \psi_1/Im(\zeta), \ |\nu_2(\zeta)| \le \psi_2/Im(\zeta) \text{ for } Im(\zeta) > C,$$

*with a $C$ sufficiently large constant.*

*Let*

$$\chi := \nu_1(i(\psi_1\psi_2\lambda)^{1/2}).\nu_2(i(\psi_1\psi_2\lambda)^{1/2}),$$

*and*

$$\begin{aligned}\mathscr{E}_0(\xi, \psi_1, \psi_2, \lambda) = &-\chi^5\xi^6 + 3\chi^4\xi^4 + (\psi_1\psi_2 - \psi_1 - \psi_2 + 1)\chi^3\xi^6 - 2\chi^3\xi^4 - 3\chi^3\xi^2 \\ &+ (\psi_1 + \psi_2 - 3\psi_1\psi_2 + 1)\chi^2\xi^4 + 2\chi^2\xi^2 + \chi^2 + 3\psi_1\psi_2\chi\xi^2 - \psi_1\psi_2,\end{aligned}$$
$$\mathscr{E}_1(\xi, \psi_1, \psi_2, \lambda) = \psi_2\chi^3\xi^4 - \psi_2\chi^2\xi^2 + \psi_1\psi_2\chi\xi^2 - \psi_1\psi_2,$$
$$\mathscr{E}_2(\xi, \psi_1, \psi_2, \lambda) = \chi^5\xi^6 - 3\chi^4\xi^4 + (\psi_1 - 1)\chi^3\xi^6 + 2\chi^3\xi^4 + 3\chi^3\xi^2 + (-\psi_1 - 1)\chi^2\xi^4 - 2\chi^2\xi^2 - \chi^2.$$

*We then define*

$$\mathscr{B}_{\text{ridge}}(\xi, \psi_1, \psi_2, \lambda) = \frac{\mathscr{E}_1(\xi, \psi_1, \psi_2, \lambda)}{\mathscr{E}_0(\xi, \psi_1, \psi_2, \lambda)}, \tag{B.1}$$

$$\mathscr{V}_{\text{ridge}}(\xi, \psi_1, \psi_2, \lambda) = \frac{\mathscr{E}_2(\xi, \psi_1, \psi_2, \lambda)}{\mathscr{E}_0(\xi, \psi_1, \psi_2, \lambda)}. \tag{B.2}$$

The quantities derived above can be easily derived numerically. But for our interest, we need to focus on the limiting case when $\lambda \to 0$. It can be shown that when $\lambda \to 0$, the expressions of $\mathscr{B}_{\text{ridge}}$ and $\mathscr{V}_{\text{ridge}}$ reduces to the form of $\mathscr{B}^\star$ and $\mathscr{V}^\star$ defined in Lemma 3.7 respectively. For details we refer to Section 5.2 in Mei and Montanari (2019a).

Now we are ready to present the proof of the main result. Recall that

$$\hat{a}(\lambda) := \arg\min_{a \in \mathbb{R}^N} \frac{1}{n} \sum_{i=1}^{n} \left( y_i - \sum_{j=1}^{N} a_j \sigma(\theta_j^\top x_i/\sqrt{d}) \right)^2 + \frac{N\lambda}{d} \|a\|_2^2.$$

By standard linear algebra it immediately follows that

$$\hat{a}(\lambda) = \frac{1}{\sqrt{d}} (Z^\top Z + \lambda\psi_{1,d}\psi_{2,d}\mathbb{I}_N)^{-1} Z^\top y, \tag{B.3}$$

where $Z = \frac{1}{\sqrt{d}}\sigma(X\Theta^\top/\sqrt{d})$, $\psi_{1,d} = \frac{N}{d}$ and $\psi_{2,d} = \frac{n}{d}$. Also, define the square error loss $R_{\mathrm{RF}}(x, X, \Theta, \lambda) := (x^\top \beta_0 - f(x; \hat{a}(\lambda); \Theta))^2$, where $x$ is a new feature point.

For brevity, we denote by $\mathbf{\Gamma}$ the tuple $(X, \Theta, \beta_0, \delta)$. Recall that we are interested in the out-of-sample risk $\mathbb{E}_{\mathbf{\Gamma}}\mathbb{E}_x[R_{\mathrm{RF}}(x, X, \Theta, \lambda)]$, where $x$ is a new feature point from minority group. The main recipe of our proof is the following:

- We first do the bias-variance decomposition of the expected out-of-sample-risk.

- We analyze the bias and variance terms separately using techniques from random matrix theory.

- Finally, we obtain asymptotic limit of the expected risk by using the asymptotic limits of bias and variance terms.

### BIAS VARIANCE DECOMPOSITION

The expected risk at a new point $x$ coming from the minority group can be decomposed into following way:

$$\mathbb{E}_{\mathbf{\Gamma}}\mathbb{E}_x[R_{\mathrm{RF}}(x, X, \Theta, \lambda)] = \mathbb{E}_{\mathbf{\Gamma}}\mathbb{E}_x\{x^\top\beta_0 - f(x; \hat{a}(\lambda); \Theta)\}^2$$
$$= \underbrace{\mathbb{E}_{\mathbf{\Gamma}}\mathbb{E}_x\left\{ \left[x^\top\beta_0 - \mathbb{E}_\epsilon f(x; \hat{a}(\lambda); \Theta)\right]^2 \right\}}_{\text{Bias term}} + \underbrace{\mathbb{E}_{\mathbf{\Gamma}}\mathbb{E}_x\mathrm{Var}_\epsilon\left\{ f(x; \hat{a}(\lambda); \Theta) \right\}}_{\text{Variance term}}. \tag{B.4}$$

The above fact follows from Lemma 3.3, which basically uses the uncorrelatedness of $\beta_0$ and $\delta$. Now we define the matrix

$$\Psi := (Z^\top Z + \lambda\psi_{1,d}\psi_{2,d}\mathbb{I}_N)^{-1}. \tag{B.5}$$

In the next couple of sections we will study the asymptotic behavior of the bias and variance terms by analyzing several terms involving $\Psi$. In order to obtain asymptotic limits of those terms, we will be borrowing results of random matrix theory in Mei and Montanari (2019a).

### DECOMPOSITION OF BIAS TERM

We focus the bias

$$\mathcal{B}(\delta, \lambda) := \mathbb{E}_{\mathbf{\Gamma}}\mathbb{E}_x\left\{ \left[x^\top\beta_0 - \mathbb{E}_\epsilon f(x; \hat{a}(\lambda); \Theta)\right]^2 \right\}$$
$$= \mathbb{E}_{\mathbf{\Gamma}}\mathbb{E}_x\left\{ \left[x^\top\beta_0 - \sigma(\Theta x/\sqrt{d})^\top \frac{1}{\sqrt{d}}\left(\Psi Z^\top X\beta_0 + \Psi Z_1^\top X_1\delta\right)\right]^2 \right\}. \tag{B.6}$$

It turns out that it is rather difficult to directly analyze the bias term $\mathcal{B}(\delta, \lambda)$. The main difficulty lies in the fact that the terms involving $\beta_0$ and $\delta$ are coupled in the bias term and poses main difficulty in applying well known random matrix theory results directly. Hence, in order to decouple the terms we look at the following decomposition:

$$\mathcal{B}(\delta, \lambda) = \underbrace{\mathcal{B}(\delta, \lambda) - \mathcal{B}(0, \lambda)}_{(I)} + \underbrace{\mathcal{B}(0, \lambda)}_{(II)}$$

It is fairly simple to see that term (II) is only a function of $\beta_0$. Next, we will show that term (I) does not contain $\beta_0$. To this end, We define the matrices $\widehat{U}, U \in \mathbb{R}^{N \times N}$ as follows:

$$\widehat{U}_{ij} = \sigma(\theta_i^\top x/\sqrt{d})\sigma(\theta_j^\top x/\sqrt{d}), \ U_{ij} = \mathbb{E}(\widehat{U}_{ij}) \ \forall (i, j) \in [N] \times [N]. \tag{B.7}$$

Next, due to Assumption 3.2, it easily follows that

$$
\begin{aligned}
&\mathcal{B}(\delta, \lambda) - \mathcal{B}(0, \lambda) \\
&= \mathbb{E}_{\mathbf{\Gamma}} \mathbb{E}_x \left\{ \left[ x^\top \beta_0 - \sigma(\Theta x / \sqrt{d})^\top \frac{1}{\sqrt{d}} \left( \Psi Z^\top X \beta_0 + \Psi Z_1^\top X_1 \delta \right) \right]^2 \right\} \\
&\quad - \mathbb{E}_{\mathbf{\Gamma}} \mathbb{E}_x \left\{ \left[ x^\top \beta_0 - \sigma(\Theta x / \sqrt{d})^\top \frac{1}{\sqrt{d}} \left( \Psi Z^\top X \beta_0 \right) \right]^2 \right\} \\
&= \frac{1}{d} \mathbb{E}_{\mathbf{\Gamma}} \mathbb{E}_x \left( \delta^\top X_1^\top Z_1 \Psi \widehat{U} \Psi Z_1^\top X_1 \delta \right) \\
&= \frac{1}{d} \mathbb{E}_{\mathbf{\Gamma}} \mathbb{E}_x \left\{ \mathrm{Tr} \left( X_1^\top Z_1 \Psi \widehat{U} \Psi Z_1^\top X_1 \delta \delta^\top \right) \right\} \\
&= \frac{F_\delta^2}{d^2} \mathbb{E} \left\{ \mathrm{Tr} \left( X_1^\top Z_1 \Psi U \Psi Z_1^\top X_1 \right) \right\}.
\end{aligned}
\tag{B.8}
$$

Above display shows that term (I) does not contain $\beta_0$. In the subsequent discussion, we will now focus on obtaining the limits of term (I) and (II) separately.

Next, by the property of trace, we have the following:

$$
\begin{aligned}
\mathrm{Tr}(X_1^\top Z_1 \Psi U \Psi Z_1^\top X_1) &= \mathrm{Tr} \left\{ \left( \sum_{i=n_0+1}^n x_i z_i^\top \right) \Psi U \Psi \left( \sum_{i=n_0+1}^n z_i x_i^\top \right) \right\} \\
&= \sum_{i=n_0+1}^{n_1} \sum_{j=n_0+1}^{n_1} \mathrm{Tr} \left( \Psi U \Psi z_j x_j^\top x_i z_i^\top \right) \\
&= \sum_{i=n_0+1}^{n_1} \sum_{j=n_0+1}^{n_1} x_j^\top x_i \mathrm{Tr} \left( \Psi U \Psi z_j z_i^\top \right) \\
&= \sum_{i=n_0+1}^n x_i^\top x_i \mathrm{Tr} \left( \Psi U \Psi z_i z_i^\top \right) + \sum_{n_0+1 \le i < j \le n} x_j^\top x_i \mathrm{Tr} \left( \Psi U \Psi z_j z_i^\top \right).
\end{aligned}
\tag{B.9}
$$

Thus, by exchangeability of the terms $\{ \mathrm{Tr} \left( \Psi U \Psi z_i z_i^\top \right) \}_{i \in [n]}$ and $\{ x_j^\top x_i \mathrm{Tr} \left( \Psi U \Psi z_j z_i^\top \right) \}_{n_0+1 \le i < j \le n}$ we have

$$
\frac{1}{d} \mathbb{E} \left\{ \mathrm{Tr} \left( X_1^\top Z_1 \Psi U \Psi Z_1^\top X_1 \right) \right\} = n_1 f(1,1) + n_1 (n_1 - 1) f(1, 2),
\tag{B.10}
$$

where $f(i, j) = \mathbb{E} \left\{ x_j^\top x_i \mathrm{Tr} \left( \Psi U \Psi z_j z_i^\top \right) \right\} / d^2$ and $1 \le i, j \le 2$. Hence, together with Equation (B.8) and Equation (B.10), it follows that

$$
\mathcal{B}(\delta, \lambda) - \mathcal{B}(0, \lambda) = F_\delta^2 \{ n_1 f(1, 1) + n_1 (n_1 - 1) f(1, 2) \}.
\tag{B.11}
$$

Furthermore, from Equation (8.10), Equation (8.25), Lemma 9.3 and Lemma 9.4 of Mei and Montanari (2019a), we get

$$
\mathcal{B}(0, \lambda) = \mathscr{B}_{\mathrm{ridge}}(\xi, \psi_1, \psi_2, \lambda / \mu_\star^2) F_\beta^2 + o_d(1),
\tag{B.12}
$$

where $\mathscr{B}_{\mathrm{ridge}}(\xi, \psi_1, \psi_2, \lambda / \mu_\star^2)$ is defined as in Equation (B.1). Thus, it only remains to calculate the limit of $\mathcal{B}(\delta, \lambda) - \mathcal{B}(0, \lambda)$ in order to obtain the limit of the bias term $\mathcal{B}(\delta, \lambda)$.

### ANALYSIS OF VARIANCE TERM

Now we briefly deffer the calculation of term (I) in Equation (B.11) and shift our focus to variance term. The reason behind this is that it is hard to directly calculate the limits of the quantities appearing in the right hand side of Equation (B.11). We obtain these limiting quantities with the help of the limiting variance.

Note that for the variance term, we have

$$
\begin{aligned}
\mathcal{V}(\lambda) &= \mathbb{E}_{\mathbf{\Gamma}} \mathbb{E}_x \mathrm{Var}_\epsilon \big\{ f(x; \hat{a}(\lambda); \Theta) \big\} \\
&= \mathbb{E}_{\mathbf{\Gamma}} \mathbb{E}_x \mathrm{Var}_\epsilon \left\{ \sigma(\Theta x/\sqrt{d})^\top \frac{1}{\sqrt{d}} \Psi Z^\top \epsilon \right\} \\
&= \frac{1}{d} \mathbb{E}_{\mathbf{\Gamma}} \mathbb{E}_x \left[ \mathrm{Tr}\big\{ \Psi \widehat{U} \Psi Z^\top Z \big\} \right] \tau^2 \\
&= \frac{\tau^2}{d} \mathbb{E} \left\{ \mathrm{Tr}\left( \Psi U \Psi Z^\top Z \right) \right\} \\
&= \widetilde{\Psi}_3 \tau^2,
\end{aligned}
\tag{B.13}
$$

where $\widetilde{\Psi}_3 := \mathbb{E}\left\{ \mathrm{Tr}\left( \Psi U \Psi Z^\top Z \right) \right\}/d$. Again by exchangeability argument, we have the following:

$$
\begin{aligned}
nf(1,1) &= \mathbb{E} \left\{ \frac{1}{d^2} \sum_{i=1}^n x_i^\top x_i \mathrm{Tr}\left( \Psi U \Psi z_i z_i^\top \right) \right\} \\
&= \mathbb{E} \left\{ \frac{1}{d} \sum_{i=1}^n \mathrm{Tr}\left( \Psi U \Psi z_i z_i^\top \right) \right\} \qquad (\text{as } x_i^\top x_i/d = 1) \\
&= \mathbb{E} \left\{ \frac{1}{d} \mathrm{Tr}\left( \Psi U \Psi Z^\top Z \right) \right\} \\
&= \widetilde{\Psi}_3.
\end{aligned}
\tag{B.14}
$$

Thus, in the light of Equation (B.11) and Equation (B.14), we reemphasize that it is essential to understand the limiting behavior of $f(i, j)$ in order to obtain the limit of expected out-of-sample risk.

OBTAINING LIMIT OF VARIANCE TERM:

To this end, we define the matrix $\mathbf{A} \in \mathbb{R}^{M \times M}$, $M = N + n$ with parameters $\mathbf{q} = (s_1, s_2, t_1, t_2, p) \in \mathbb{R}^5$ in the following way:

$$
\mathbf{A} = \mathbf{A}(\mathbf{q}) := \begin{bmatrix} s_1 \mathbf{I}_N + s_2 \mathbf{Q} & Z^\top + p\mathbf{W}^\top \\ Z + p\mathbf{W} & t_1 \mathbf{I}_n + t_2 \mathbf{H} \end{bmatrix}
$$

where

$$
\mathbf{Q} = \frac{1}{d} \Theta \Theta^\top, \mathbf{H} = \frac{1}{d} X X^\top, \mathbf{W} = \frac{\mu_1}{d} X \Theta^\top.
$$

For $\xi \in \mathbb{C}_+$, we define the log-determinant of $\mathbf{A}$ as $G(\xi, \mathbf{q}) = \left[ \frac{1}{d} \sum_{i=1}^M \mathrm{Log}\left( \lambda_i(\mathbf{A}) - \xi \mathbf{I}_M \right) \right]$. Here Log is the branch cut on the negative real axis and $\{\lambda_i(\mathbf{A})\}_{i \in [M]}$ denotes the eigenvalues of $\mathbf{A}$ in non-increasing order.

Define the quantity

$$
\breve{\Psi}_3 := \frac{1}{d} \mathrm{Tr}\big( \Psi U \Psi Z^\top Z \big).
$$

From Equation (B.14), it trivially follows that $\mathbb{E}(\breve{\Psi}_3) = nf(1,1)$. The key trick lies in replacing the kernel matrix $U$ by the matrix $\mathbf{\Lambda} = \mu_1^2 \mathbf{Q} + \mu_\star^2 \mathbf{I}_N$. By (Mei and Montanari, 2019a, Lemma 9.4), we know that the asymptotic error incurred in the expected value of $\breve{\Psi}_3$ by replacing $U$ in place of $\mathbf{\Lambda}$ is $o_d(1)$. To elaborate, we first define

$$
\Psi_3 := \frac{1}{d} \mathrm{Tr}\big( \Psi \mathbf{\Lambda} \Psi Z^\top Z \big).
$$

Then by (Mei and Montanari, 2019a, Lemma 9.4), we have

$$
\mathbb{E}|\breve{\Psi}_3 - \Psi_3| = o_d(1).
\tag{B.15}
$$

Thus, instead of $\breve{\Psi}_3$, we will focus on $\Psi_3$. By (Mei and Montanari, 2019a, Proposition 8.2) we know

$$
\Psi_3 = -\mu_\star^2 \partial_{s_1, t_1} G_d(i(\psi_1 \psi_2 \lambda)^{1/2}; \mathbf{0}) - \mu_1^2 \partial_{s_2, t_1} G_d(i(\psi_1 \psi_2 \lambda)^{1/2}; \mathbf{0}).
\tag{B.16}
$$

Also by (Mei and Montanari, 2019a, Proposition 8.5), for any fixed $u \in \mathbb{R}_+$, we have the following:

$$\lim_{d \to \infty} \mathbb{E}\left[\left\|\nabla_{\mathbf{q}}^2 G_d(iu; \mathbf{0}) - \nabla_{\mathbf{q}}^2 g(iu; \mathbf{0})\right\|_{op}\right] = 0. \tag{B.17}$$

As $\Psi_3$ is a bi-linear form of $\nabla_{\mathbf{q}}^2 G_d(i(\psi_1 \psi_2 \lambda)^{1/2}; \mathbf{0})$ (See Equation (B.16)), together with (Mei and Montanari, 2019a, Equation (8.26)), it follows that

$$\lim_{d \to \infty} \mathbb{E}\left|\Psi_3 - \mathscr{V}_{\mathrm{ridge}}(\xi, \psi_1, \psi_2, \lambda/\mu_\star^2)\right| \to 0, \tag{B.18}$$

where $\mathscr{V}_{\mathrm{ridge}}(\xi, \psi_1, \psi_2, \lambda/\mu_\star^2)$ is as defined in Equation (B.2). Thus, both Equation (B.15) and Equation (B.18) yields that

$$\lim_{d \to \infty} \mathbb{E}|\breve{\Psi}_3 - \mathscr{V}_{\mathrm{ridge}}(\xi, \psi_1, \psi_2, \lambda/\mu_\star^2)| \to 0. \tag{B.19}$$

Hence, by Equation (B.13) we get the following:

$$\mathcal{V}(\lambda) = \mathscr{V}_{\mathrm{ridge}}(\xi, \psi_1, \psi_2, \lambda/\mu_\star^2)\tau^2 + o_d(1). \tag{B.20}$$

OBTAINING LIMIT OF BIAS TERM:

Now we revisit the term $\mathcal{B}(\delta, \lambda) - \mathcal{B}(0, \lambda)$. We will study the terms $n_1 f(1, 1)$ and $n_1(n_1 - 1)f(1, 2)$ in Equation (B.11) separately.

First, we focus on the term $n_1 f(1, 1)$. Equation (B.14), Equation (B.15) and Equation (B.19) show that

$$nf(1, 1) = \mathscr{V}_{\mathrm{ridge}}(\xi, \psi_1, \psi_2, \lambda/\mu_\star^2) + o_d(1). \tag{B.21}$$

Next, in order to understand the limiting behavior of $f(1, 2)$, we define

$$\breve{\Psi}_2 := \frac{1}{d}\mathrm{Tr}\left(\Psi U \Psi Z^\top \mathbf{H} Z\right), \quad \Psi_2 := \frac{1}{d}\mathrm{Tr}\left(\Psi \mathbf{\Lambda} \Psi Z^\top \mathbf{H} Z\right).$$

Again due to (Mei and Montanari, 2019a, Lemma 9.4), we have

$$\mathbb{E}|\breve{\Psi}_3 - \Psi_3| = o_d(1).$$

Hence, we only focus on $\Psi_2$. A calculation similar to (B.9) shows that

$$\mathbb{E}(\Psi_2) = nf(1, 1) + n(n - 1)f(1, 2) + o_d(1). \tag{B.22}$$

Let $g(\xi; \mathbf{q})$ be the analytic function defined in (Mei and Montanari, 2019a, Equation (8.19)). Using (Mei and Montanari, 2019a, Proposition 8.5), we get

$$\mathbb{E}(\Psi_2) = -\mu_\star^2 \partial_{s_1, t_2} g(i(\psi_1 \psi_2 \lambda)^{1/2}; \mathbf{0}) - \mu_1^2 \partial_{s_2, t_2} g(i(\psi_1 \psi_2 \lambda)^{1/2}; \mathbf{0}) + o_d(1) =: \Psi_2^\star(\xi, \psi_1, \psi_2, \lambda, \mu_\star, \mu_1) + o_d 1.$$

This along with (B.21) and (B.22) show that

$$n(n - 1)f(1, 2) = \Psi_2^\star(\xi, \psi_1, \psi_2, \lambda, \mu_\star, \mu_1) - \mathscr{V}_{\mathrm{ridge}}(\xi, \psi_1, \psi_2, \lambda/\mu_\star^2) + o_d(1). \tag{B.23}$$

Thus, finally using Equation (B.11), Equation (B.12) and the fact $n_1/n \to \pi$, we conclude that

$$\mathcal{B}(\delta, \lambda) = F_\beta^2 \mathscr{B}_{\mathrm{ridge}}(\xi, \psi_1, \psi_2, \lambda/\mu_\star^2) + F_\delta^2[\pi(1 - \pi)\mathscr{V}_{\mathrm{ridge}}(\xi, \psi_1, \psi_2, \lambda/\mu_\star^2) + \pi^2 \Psi_2^\star(\xi, \psi_1, \psi_2, \lambda, \mu_\star, \mu_1)] + o_d(1). \tag{B.24}$$

Lastly, using Equation (B.20), Equation (B.24) and plugging the values in Equation (B.4) we get

$$\lim_{d \to \infty} \mathbb{E}_{X, \Theta, \beta_0, \delta}[R_{\mathrm{RF}}(x, X, \Theta, \lambda)]$$
$$= F_\beta^2 \mathscr{B}_{\mathrm{ridge}}(\xi, \psi_1, \psi_2, \lambda/\mu_\star^2) + F_\delta^2[\pi(1 - \pi)\mathscr{V}_{\mathrm{ridge}}(\xi, \psi_1, \psi_2, \lambda/\mu_\star^2) + \pi^2 \Psi_2^\star(\xi, \psi_1, \psi_2, \lambda, \mu_\star, \mu_1)]$$
$$+ \tau^2 \mathscr{V}_{\mathrm{ridge}}(\xi, \psi_1, \psi_2, \lambda/\mu_\star^2). \tag{B.25}$$

RIDGELESS LIMIT

Finally, for ridgeless limit we need to take $\lambda \to 0+$ on both sides of Equation (B.25). Following similar calculations as in the proof of (Mei and Montanari, 2019a, Theorem 5.7), or more specifically using (Mei and Montanari, 2019a, Lemma 12.1), we ultimately get

$$\lim_{\lambda \to 0+} \lim_{d \to \infty} \mathbb{E}_{X, \Theta, \beta_0, \delta}[R_{\mathrm{RF}}(x, X, \Theta, \lambda)] = F_\beta^2 \mathscr{B}^\star + F_\delta^2 \mathscr{M}_1^\star + \tau^2 \mathscr{V}^\star.$$

This completes the proof.

## B.2 Bias-variance decomposition under orthogonality of parameters

In this section we will demonstrate that the same bias-variance decomposition in Lemma 3.3 continues to hold rather weaker assumption than Assumption 3.2. Specifically, in thi section we only assume the parameter vectors $\beta_0$ and $\delta$ are orthogonal, i.e., $\beta_0^\top \delta = 0$. The following Lemma shows that under this orthogonality condition, the desired bias-variance decomposition still holds.

**Lemma B.2.** *Define the followings:* $\mathscr{B}_\beta = \mathbb{E}_{x \sim P_X}[\|z^\top (Z^\top Z)^\dagger Z^\top X - x^\top\|_2^2/d]$ *and* $\mathscr{B}_\delta = \mathbb{E}_{x \sim P_X}[\|z^\top (Z^\top Z)^\dagger Z_1^\top X_1\|_2^2/d]$. *Also assume that* $\beta_0^\top \delta = 0$. *Then, we have* $\mathscr{B}(\beta_0, \delta) = F_\beta^2 \mathscr{B}_\beta + F_\delta^2 \mathscr{B}_\delta$.

*Proof.* Similar to Equation (B.4), her also we have

$$\mathscr{B}(\beta_0, \delta) = \mathbb{E}_{x \sim P_X}[\{(z^\top (Z^\top Z)^\dagger Z^\top X - x^\top)\beta_0\}^2 + \{z^\top (Z^\top Z)^\dagger Z_1^\top X_1 \delta\}^2$$
$$+ 2(z^\top (Z^\top Z)^\dagger Z^\top X - x^\top)\beta_0 \delta^\top X_1^\top Z_1 (Z^\top Z)^\dagger z]. \tag{B.26}$$

The only difference in Equation (B.4) compare to Equation (B.26) is that $\beta_0$ and $\delta$ are now fixed vectors in $\mathbb{R}^d$ instead of being random. To this end we define the space of $d \times d$ orthogonal matrices as follows:

$$\mathcal{O}(d) := \{O \in \mathbb{R}^{d \times d} : O^\top O = OO^\top = \mathbb{I}_d\}.$$

Now we consider the following change of variables for a matrix $O \in \mathcal{O}(d)$:

$$X \mapsto XO^\top \triangleq \bar{X}, \quad \Theta \mapsto \Theta O^\top \triangleq \bar{\Theta}$$
$$\beta_0 \mapsto O\beta_0 \triangleq \bar{\beta}_0, \quad \delta \mapsto O\delta \triangleq \bar{\delta}, \quad x \mapsto Ox \triangleq \bar{x}. \tag{B.27}$$

The key thing here is to note the following

$$\bar{Z} \triangleq \sigma(\bar{X}\bar{\Theta}^\top/\sqrt{d})/\sqrt{d} = \sigma(X\Theta^\top/\sqrt{d})/\sqrt{d} = Z.$$

Thus, the expression of $\hat{a}(\lambda)$ in Equation (B.3) remains unchanged. Finally, due to distributional assumption on $x$ and Lemma C.1, we have $\bar{x} \sim \text{Unif}\{\mathbb{S}^{d-1}(\sqrt{d})\}$ for every $O \in \mathcal{O}(d)$. Finally, note that $\bar{x}^\top \bar{\beta}_0 = x^\top \beta$, which entails that $\mathscr{B}(\beta_0, \delta) = \bar{\mathscr{B}}(\bar{\beta}_0, \bar{\delta})$ for all $O \in \mathcal{O}(d)$, where

$$\bar{\mathscr{B}}(\bar{\beta}_0, \bar{\delta}) = \mathbb{E}_{\bar{x} \sim P_X}[\{(\bar{z}^\top (\bar{Z}^\top \bar{Z})^\dagger \bar{Z}^\top \bar{X} - \bar{x}^\top)\bar{\beta}_0\}^2 + \{\bar{z}^\top (\bar{Z}^\top \bar{Z})^\dagger \bar{Z}_1^\top \bar{X}_1 \bar{\delta}\}^2$$
$$+ 2(\bar{z}^\top (\bar{Z}^\top \bar{Z})^\dagger \bar{Z}^\top \bar{X} - \bar{x}^\top)\bar{\beta}_0 \bar{\delta}^\top \bar{X}_1^\top \bar{Z}_1 (\bar{Z}^\top \bar{Z})^\dagger \bar{z}]$$

This motivates us to consider the change of variables in Equation (B.27), when $O$ is sampled from Haar measure $\mathcal{H}_d$ on $\mathcal{O}(d)$ and independent of $X, \Theta, \epsilon$. Also, note that due to Lemma C.1 and Lemma C.2, the random vectors $\bar{\beta}_0$ and $\bar{\delta}$ satisfy the setup of Section B.1. Thus the result follows immediately by noting the fact that

$$\mathscr{B}(\beta_0, \delta) = \mathbb{E}_{O \sim \mathcal{H}_d}[\bar{\mathscr{B}}(\bar{\beta}_0, \bar{\delta})] = F_\beta^2 \mathscr{B}_\beta + F_\delta^2 \mathscr{B}_\delta,$$

where the last inequality follows from taking $\lambda \to 0$ in Equation (B.11) and Equation (B.12).

$\square$

## B.3 Limiting risk for general $\beta_0$ and $\delta$

Unlike the previous section, this section studies the general property of the asymptotic risk $R_0(\hat{a})$ when the parameters $\beta_0$ and $\delta$ are in general position. Specifically, in this section we relax the assumption that $\beta_0^\top \delta \neq 0$. Thus, the only difficulty arise in analysing the cross-covariance term $2\mathbb{E}_{x \sim P_X}[(z^\top (Z^\top Z)^\dagger Z^\top X/\sqrt{d} - x^\top)\beta_0 \delta^\top X_1^\top Z_1 (Z^\top Z)^\dagger z/\sqrt{d}]$ in Equation (3.3) as it does not vanish under the absence of orthogonality assumption. Thus, essentially it boils down to showing the results in Lemma 3.3 and Lemma 3.8.

### B.3.1 Proof of Lemma 3.3

We begin with proof of Lemma 3.3. We recall the right Equation (3.3), i.e,

$$\mathscr{B}(\beta_0, \delta) = \mathbb{E}_{x \sim P_X}[\{(z^\top (Z^\top Z)^\dagger Z^\top X - x^\top)\beta_0\}^2 + \{z^\top (Z^\top Z)^\dagger Z_1^\top X_1 \delta\}^2$$
$$+ 2(z^\top (Z^\top Z)^\dagger Z^\top X - x^\top)\beta_0 \delta^\top X_1^\top Z_1 (Z^\top Z)^\dagger z].$$

Apart from the cross-correlation term, The first two-terms of the equation does not depend on the interaction between $\beta_0$ and $\delta$, To be precise, the first two terms are only the functions of $\beta_0$ and $\delta$ individually and does not depend on the orthogonality of $\beta_0$ and $\delta$. Thus, an analysis similar to Section B.2 yields that

$$\mathscr{B}(\beta_0, \delta) = F_\beta^2 \mathscr{B}_\beta + F_\delta^2 \mathscr{B}_\delta + 2(z^\top (Z^\top Z)^\dagger Z^\top X - x^\top)\beta_0 \delta^\top X_1^\top Z_1 (Z^\top Z)^\dagger z].$$

Thus, it only remains to analyze the cross-covariance term

$$\tilde{\mathscr{C}}_{\beta,\delta} \triangleq 2\mathbb{E}_{x \sim P_X}[(z^\top (Z^\top Z)^\dagger Z^\top X - x^\top)\beta_0 \delta^\top X_1^\top Z_1 (Z^\top Z)^\dagger z],$$

To get the desired form as in Lemma 3.3, we again introduce that random variables $\bar{\beta}_0, \bar{\delta}, \bar{X}, \bar{X}_1, \bar{Z}, \bar{Z}_1$ and $\bar{z}$ as in Section B.2. Also, define the quantity

$$\bar{\mathscr{C}}_{\bar{\beta},\bar{\delta}} \triangleq 2\mathbb{E}_{\bar{x} \sim P_X}[(\bar{z}^\top (\bar{Z}^\top \bar{Z})^\dagger \bar{Z}^\top \bar{X} - \bar{x}^\top)\bar{\beta}_0 \bar{\delta}^\top \bar{X}_1^\top \bar{Z}_1 (\bar{Z}^\top \bar{Z})^\dagger \bar{z}].$$

Following, a similar argument as in Section B.2, we have $\tilde{\mathscr{C}}_{\beta,\delta} = \mathbb{E}_{O \sim \mathcal{H}_d}\{\bar{\mathscr{C}}_{\bar{\beta},\bar{\delta}}\}$. Next, note that,

$$\tilde{\mathscr{C}}_{\beta,\delta} = 2\mathbb{E}[\mathrm{Tr}(\bar{X}_1^\top \bar{Z}_1 (\bar{Z}^\top \bar{Z})^\dagger \bar{z}(\bar{z}^\top (\bar{Z}^\top \bar{Z})^\dagger \bar{Z}^\top \bar{X} - \bar{x}^\top)\bar{\beta}_0 \bar{\delta}^\top)].$$

Thus condition on $\bar{x}$ and $\bar{z}$ and applying Lemma C.2, the result follows, i.e.,

$$\tilde{\mathscr{C}}_{\beta,\delta} = F_{\beta,\delta}\mathscr{C}_{\beta,\delta}. \tag{B.28}$$

### B.3.2 Proof of Lemma 3.8

Following the same arguments of Section B.1 and Section B.2, it can be shown that the limiting values of $\mathbb{E}[\mathscr{B}_\beta]$, $\mathbb{E}[\mathscr{B}_\delta]$ and $\mathbb{E}[\mathscr{V}(\tau)]$ remains the same, i.e., the results of Lemma 3.7 and first part of Lemma 3.8 remains true. Thus, it only remains to analyze the cross-covariance term

$$\tilde{\mathscr{C}}_{\beta,\delta} \triangleq 2\mathbb{E}_{x \sim P_X}[(z^\top (Z^\top Z)^\dagger Z^\top X - x^\top)\beta_0 \delta^\top X_1^\top Z_1 (Z^\top Z)^\dagger z],$$

to fully characterize the asymptotic risk asymptotically. To begin with, the vector $\delta$ can be written as a direct sum of two components $\delta_1$ and $\delta_2$, where $\delta_2$ is the orthogonal projection of $\delta$ on $\mathrm{span}(\{\beta_0\})$. In particular we have

$$\delta = \delta_1 + \delta_2, \quad \text{and} \quad \delta_2 = (\mathbb{I}_d - \beta_0 \beta_0^\top / \|\beta_0\|^2)\delta.$$

Thus, there exists $\mu \in \mathbb{R}$ such that $\delta_1 = \mu\beta_0$. Thus, $\tilde{\mathscr{C}}_{\beta,\delta}$ can be decomposed in the following way:

$$\tilde{\mathscr{C}}_{\beta,\delta} = \underbrace{2\mu\mathbb{E}_{x \sim P_X}[\{\beta_0^\top X^\top Z(Z^\top Z)^\dagger zz^\top (Z^\top Z)^\dagger Z_1^\top X_1 \beta_0]}_{\mathscr{C}_{\beta,\delta}^{(1,1)}} - \underbrace{2\mu\mathbb{E}_{x \sim P_X}[\{\beta_0^\top X_1^\top Z_1 (Z^\top Z)^\dagger zx^\top \beta_0]}_{\mathscr{C}_{\beta,\delta}^{(1,2)}}$$

$$+ \underbrace{2\mathbb{E}_{x \sim P_X}[(z^\top (Z^\top Z)^\dagger Z^\top X - x^\top)\beta_0 \delta_2^\top X_1^\top Z_1 (Z^\top Z)^\dagger z]}_{\mathscr{C}_{\beta,\delta}^{(2)}}.$$

As $\beta_0^\top \delta_2 = 0$, following the arguments in Section B.2 we have $\mathscr{C}_{\beta,\delta}^{(2)} = 0$. To analyze the terms $\mathscr{C}_{\beta,\delta}^{(1,1)}$ and $\mathscr{C}_{\beta,\delta}^{(1,1)}$, we will gain analyze their corresponding ridge equivalents

$$\mathscr{C}_{\beta,\delta}^{(1,1)}(\lambda) := \frac{2\mu}{d}\mathbb{E}_x \left( \beta_0^\top X^\top Z\Psi\widehat{U}\Psi Z_1^\top X_1 \beta_0 \right),$$

$$\mathscr{C}_{\beta,\delta}^{(1,2)}(\lambda) := \frac{2\mu}{\sqrt{d}}\mathbb{E}_x \left( \beta_0^\top X_1^\top Z_1 \Psi u x^\top \beta_0 \right),$$

where $\Psi$ and $\widehat{U}$ are defined in Equation (B.5) and Equation (B.7) respectively and $u = \sqrt{d}z$. We focus on these terms separately. To start with, by an application of exchangeability argument we note that

$$\mathbb{E}(\mathscr{C}_{\beta,\delta}^{(1,1)}(\lambda)) = \frac{2\mu F_\beta^2}{d^2}\mathbb{E}\left[\mathrm{Tr}\left\{X^\top Z\Psi U\Psi Z_1 X_1\right\}\right]$$

$$= 2\mu F_\beta^2[n_1 f(1,1) + n_1(n_1 - 1)f(1,2) + n_0 n_1 f(1,2)].$$

Using Equation (B.21) and Equation (B.23) it can be deduced that

$$\mathbb{E}(\mathscr{C}_{\beta,\delta}^{(1,1)}(\lambda)) = 2\mu F_\beta^2 \pi \Psi_2^\star(\xi, \psi_1, \psi_2, \lambda, \mu_\star, \mu_1) + o_d(1). \tag{B.29}$$

Let us denote by $V$ the matrix $\mathbb{E}(ux^\top)$. This shows that

$$\mathbb{E}(\mathscr{C}_{\beta,\delta}^{(1,2)}(\lambda)) = \frac{2\mu}{\sqrt{d}} \mathbb{E}\left(\beta_0^\top X_1^\top Z_1 \Psi V \beta_0\right).$$

Again by exchangeability argument we get

$$\mathbb{E}(\mathscr{C}_{\beta,\delta}^{(1,2)}(\lambda)) = \left(\frac{n_1}{n}\right) \cdot \frac{2\mu}{\sqrt{d}} \mathbb{E}\left(\beta_0^\top X^\top Z \Psi V \beta_0\right).$$

Let use define $T_1 := \frac{1}{\sqrt{d}}\mathbb{E}\left(\beta_0^\top X^\top Z \Psi V \beta_0\right)$. By the arguments of Section 9.1 in Mei and Montanari (2019a), we can also conclude that

$$T_1 = \frac{F_\beta^2}{2} \partial_p g(i(\psi_1 \psi_2 \lambda)^{1/2}; \mathbf{0}) + o_d(1),$$

the function is defined in (Mei and Montanari, 2019a, Equation (8.19)). Thus we have $\mathbb{E}(\mathscr{C}_{\beta,\delta}^{(1,2)}(\lambda)) = \mu F_\beta^2 \pi \partial_p g(i(\psi_1 \psi_2 \lambda)^{1/2}; \mathbf{0}) + o_d(1)$. This along with Equation (B.29) yields the following:

$$\begin{aligned}
\mathbb{E}(\tilde{\mathscr{C}}_{\beta,\delta}) &= \lim_{\lambda \to 0} \mathbb{E}(\mathscr{C}_{\beta,\delta}^{(1,1)}(\lambda)) + \mathbb{E}(\mathscr{C}_{\beta,\delta}^{(1,2)}(\lambda)) + o_d(1) \\
&= \lim_{\lambda \to 0} \mu \pi F_\beta^2 [2\Psi_2^\star(\xi, \psi_1, \psi_2, \lambda, \mu_\star, \mu_1) - \partial_p g(i(\psi_1 \psi_2 \lambda)^{1/2}; \mathbf{0})] + o_d(1) \\
&= \mu \pi F_\beta^2 (\mathscr{B}^\star - 1 + \Psi_2^\star) + o_d(1).
\end{aligned}$$

Finally, by a simple algebra it follows that $\mu = \frac{F_\delta}{F_\beta} \cos \phi_{\beta,\delta}$, where

$$\phi_{\beta,\delta} = \arccos\left(\frac{\langle \beta, \delta \rangle}{F_\beta F_\delta}\right).$$

This shows that $\mu F_\beta^2 = F_\beta F_\delta \cos(\phi_{\beta,\delta}) = \beta_0^\top \delta = F_{\beta,\delta}$. Now recalling Equation (B.28) we have $\lim_{d \to \infty} \mathbb{E}[\mathscr{C}_{\beta,\delta}] = \pi(\mathscr{B}^\star - 1 + \Psi_2^\star)$. This finished teh proof of Lemma 3.8.

## B.4 FINAL FORM OF LIMITING RISK

Finally, Gathering all the results from Section B.1, B.2 and B.3, we have

$$\lim_{d \to \infty} \mathbb{E}(R_0(\hat{a})) = F_\beta^2 \mathscr{B}^\star + F_\delta^2 \mathscr{M}_1^\star + F_{\beta,\delta} \mathscr{M}_2^\star + \tau^2 \mathscr{V}^\star.$$

with $\mathscr{C}^\star = \pi(\mathscr{B}^\star - 1 + \Psi_2^\star)$. This concludes the proof of Theorem 3.9.

## C UNIFORM DISTRIBUTION ON SPHERE AND HAAR MEASURE

In this section we will discuss some useful results related to Uniform distribution on sphere and the Haar measure $\mathcal{H}_d$ on $\mathcal{O}(d)$.

**Lemma C.1.** *Let $U \sim Unif\{\mathbb{S}^{d-1}(\sqrt{d})\}$ and $O \in \mathcal{O}(d)$. Then $OU \sim Unif\{\mathbb{S}^{d-1}(\sqrt{d})\}$. Also, if $O \sim \mathcal{H}_d$ and is independent of $U$, then also $OU \sim Unif\{\mathbb{S}^{d-1}(\sqrt{d})\}$.*

*Proof.* Note that $U \stackrel{d}{=} \sqrt{d}G/\|G\|_2$, where $G \sim \mathbb{N}(0, \mathbb{I}_d)$. Next define $\tilde{G} := OG$. By property of Gaussian random vector we have $\tilde{G} \stackrel{d}{=} G$. Now the result follows from the following:

$$OU \stackrel{d}{=} \sqrt{d}\frac{OG}{\|G\|_2} = \sqrt{d}\frac{\tilde{G}}{\|\tilde{G}\|_2} \stackrel{d}{=} \sqrt{d}\frac{G}{\|G\|_2} \stackrel{d}{=} U.$$

Next, let $O \sim \mathcal{H}_d$ be an independent random matrix from $U$. Thus conditional on $O$, we have

$$OU \mid O \sim \text{Unif}\{\mathbb{S}^{d-1}(\sqrt{d})\}.$$

Thus, unconditionally we have $OU \sim \text{Unif}\{\mathbb{S}^{d-1}(\sqrt{d})\}$. □

**Lemma C.2.** *Let* $\beta, \delta \in \mathbb{R}^d$ *such that* $\beta^\top \delta = F_\beta, \delta$. *Also, define the random vectors* $\bar{\beta} = O\beta$ *and* $\bar{\delta} = O\delta$, *where* $O \sim \mathcal{H}_d$. *Then the followings are true:*

$$\mathbb{E}(\bar{\beta}\bar{\delta}^\top) = \frac{F_{\beta,\delta}}{d}\mathbb{I}_d$$

$$\mathbb{E}(\bar{\beta}\bar{\beta}^\top)/\|\beta\|^2 = \mathbb{E}(\bar{\delta}\bar{\delta}^\top)/\|\delta\|^2 = \frac{1}{d}\mathbb{I}_d.$$

*Proof.* Let $O = (o_1, o_2, \ldots, o_d)^\top$, i.e, $o_i$ is the $i$th row of $O$. Let $\mathbf{T} = \mathbb{E}(\bar{\beta}\bar{\delta}^\top) = \mathbb{E}(O\beta\delta^\top O^\top)$. Also, by property of Haar measure, we have $\Pi O \overset{d}{=} O$ for any permutation matrix $\Pi$. This shows that $\{o_i\}_{i=1}^d$ are exchangeable. As a consequence we have

$$\mathbb{E}(o_i o_i^\top) = \frac{\sum_{k=1}^d \mathbb{E}(o_k o_k^\top)}{d} = \frac{\mathbb{E}(O^\top O)}{d} = \frac{1}{d}\mathbb{I}_d \quad \text{for all } i \in [d].$$

Now we are equipped to compute matrix $\mathbf{T}$. Note that

$$\mathbf{T}_{ii} = \mathbb{E}(o_i^\top \beta \delta^\top o_i) = \mathbb{E}\{\delta^\top o_i o_i^\top \beta\} = \delta^\top \beta/d = F_{\beta,\delta}/d.$$

Also, for $i \neq j$, we similarly get

$$\mathbf{T}_{ij} = \mathbb{E}\{\delta^\top o_j o_i^\top \beta\}.$$

Now again by property of Haar measure we know $O \overset{d}{=} (\mathbb{I}_d - 2e_i e_i^\top)O$, where $e_i$ denotes the $i$th canonical basis of $\mathbb{R}^d$. This implies that $o_j o_i^\top \overset{d}{=} -o_j o_i^\top$. Using, this property of $\mathcal{H}_d$, we also get $\mathbb{E}(o_j o_i^\top) = 0$. Thus, we get $\mathbf{T}_{ij} = 0$ and this concludes the proof for uncorrelatedness.

Next, we define $\mathbf{V}_\beta := \mathbb{E}(\bar{\beta}\bar{\beta}^\top)$. By a similar argument, it also follows that

$$(\mathbf{V}_\beta)_{i,j} = \frac{\|\beta\|_2^2}{d}\Delta_{ij},$$

where $\Delta_{ij}$ is the Kronecker delta function. The result for $\bar{\delta}$ can be shown following exactly the same recipe. □

## D  RANDOM FEATURE CLASSIFICATION MODEL

In this section, we consider an overparameterized classification problem. We consider a two group setup: $g \in \{0, 1\}$. Without loss of generality, we assume $\pi > \frac{1}{2}$ (so the $g = 1$ group is the majority group). The data generating process of the training examples $\{(x_i, y_i)\}_{i=1}^n$ is

$$
\begin{aligned}
g_i &\sim \text{Ber}(\pi) \\
x_i &\sim \mathcal{N}(0, I) \\
y_i | x_i, g_i &\leftarrow \begin{cases} +1, & \text{w.p. } f(x_i^\top \beta_0)\mathbf{1}\{g_i = 0\} + f(x_i^\top \beta_1)\mathbf{1}\{g_i = 1\} \\ -1, & \text{w.p. } 1 - f(x_i^\top \beta_0)\mathbf{1}\{g_i = 0\} - f(x_i^\top \beta_1)\mathbf{1}\{g_i = 1\} \end{cases}
\end{aligned}
\tag{D.1}
$$

where $\beta_0, \beta_1 \in \mathbf{R}^d$ are vectors of coefficients of the minority and majority groups respectively, and $f(t) = (1 + e^{-t})^{-1}$ is the sigmoid function.

Consider a random feature classification model, that is, for a newly generated sample $(x_{n+1}, y_{n+1})$, the classifier which predicts a label $\hat{y}_{n+1}$ for the new sample as

$$\hat{y} = \text{sign}\left(\sum_{j=1}^N a_j \sigma\left(\theta_j^\top x_{n+1}/\sqrt{d}\right)\right),$$

where $\sigma(\cdot)$ is a non-linear activation function and $N$ is the number of random features considered in the model. In the overparameterized setting ($n \leq N$), we train the random feature classification model by solving the following hard-margin SVM problem:

$$\widehat{a} \in \begin{cases} \arg\min_{a \in \mathbf{R}^N} & \|a\|_2 \\ \text{subject to} & y_i z_i^\top a \geq 1, \forall i \in [n] \end{cases},$$

where $z_i = (z_{i,1}, \ldots, z_{i,N})^\top$ with $z_{i,j} = \sigma(\theta_j^\top x_i / \sqrt{d})$ for $i \in [n], j \in [N]$.

We wish to study disparity between the asymptotic risks (*i.e.*, test-time classification errors) of $\widehat{a}$ on the minority and majority groups, that is, comparing

$$\mathcal{R}_0(\widehat{a}) = \mathbf{E}[\mathbf{1}\{\hat{y}_{n+1} \neq y_{n+1}\}|g_{n+1} = 0] \quad \text{and} \quad \mathcal{R}_1(\widehat{a}) = \mathbf{E}[\mathbf{1}\{\hat{y}_{n+1} \neq y_{n+1}\}|g_{n+1} = 1]$$

asymptotically. Moreover, we consider a high-dimensional asymptotic regime such that

$$d \to \infty, N/d \to \psi_1 > 0, n/d \to \psi_2 > 0 \quad \text{and} \quad \gamma = \psi_1/\psi_2 = \lim_{d \to \infty}\{N/n\} \geq 1.$$

Therefore, $\gamma$ encodes the level of overparameterization.

In the simulation for Figure 3, we let $\sigma(\cdot)$ be the ReLU activation function and $\theta_{i,j}$ be IID standard normal distributed. Moreover, we let $\pi = 0.95$, $\beta_0 = 10e_1$, $\beta_1 = 10\cos(\theta)e_1 + 10\sin(\theta)e_2$, $n = 400, d = 200, N = \gamma n$ where $e_1$ and $e_2$ are the first two standard basis of $\mathbf{R}^d$. We tune hyperparameters $\theta \in \{0°, 45°, 90°, 135°, 180°\}$ and $\gamma \in \{1, 1.5, 2, 2.5, 3, 3.5, 4, 4.5, 5, 5.5, 6\}$, then report test errors averaged over 20 replicates.

