# OpenReview forum: "How does overparametrization affect performance on minority groups?"
_ICLR.cc/2023/Conference — Submitted to ICLR 2023_

### Official Review · Reviewer_2EGb · 2022-10-24

**Confidence:** 1
**Correctness:** 3
**Technical Novelty And Significance:** 2
**Empirical Novelty And Significance:** 2
**Recommendation:** 5

**Clarity, Quality, Novelty And Reproducibility:**

It provides a theoretical justification for the empirical results on the overparameterazation on minority group

**Strength And Weaknesses:**

It provides a theoretical justification for the effect of overparameterization on minority groups.
The paper is easy to follow.

The application scenario of overparameterization and underparameterization in machine learning is not well explained.

It mentions that the proposed two-group model has parameters for controlling signal strength, majority group fraction and many terms. I would like to see how the signal strength is used in classification, regression or other machine learning task.


**Summary Of The Paper:**

This paper develops a simple two-group model to study the effects of overparameterization on groups. It provides a theoretical justification for the empirical results in the existing work and shows that the overparameterization improves or does not harm the minority risk of ERM. The majority group subsampling improves minority group performance in the overparameterization regime.

**Summary Of The Review:**

see above.

---

### Official Review · Reviewer_BTHL · 2022-10-26

**Confidence:** 4
**Correctness:** 4
**Technical Novelty And Significance:** 1
**Empirical Novelty And Significance:** Not applicable
**Recommendation:** 3

**Clarity, Quality, Novelty And Reproducibility:**

The writing of this works needs to be improved. Some of the statements are unclear in the paper.

On page 3, this work says that the minority group error increases with overparameterization in the linear model setting but later in the appendix, it says that even the average error increases in this setting.

One page 1, this works says that [1] showed that overparameterization hurts minority group accuracy whereas at multiple points, they claim in the paper that confirm [1]’s findings by showing that overparameterization helps minority group accuracy. Can the authors please clarify this?

**Strength And Weaknesses:**

The problem that this work studies of how overparameterization affects subpopulations is important for the fairness and safety of machine learning and has gained widespread interest.

I think that the model that this work studies is different from the practical setting that this work motivates. In particular, this work assumes that the subgroups have same feature distribution and only differ in their true conditional output distribution. I don’t see why that assumption is correct because the different subgroups would have different feature distribution. In fact, they assume that the two groups have different output distribution conditional on the features which is not clear if it is true because we can think of both the populations having the same true predictor. The setting that this work considers there does not exist a single good predictor which is not true in the empirical papers that they cite. This is also what was considered in the original paper [1] which first studied this using a toy model.

The paper misses important citations and comparisons to previous works which have studied similar problems [2,3,4 ].

[1] An investigation of why overparameterization exacerbates spurious correlations
[2] Covariate Shift in High-Dimensional Random Feature Regression
[3] Undersampling is a minimax optimal robustness intervention in nonparametric classification
[4] Throwing away data improves worst-class error in imbalanced classification

**Summary Of The Paper:**

Recently, it has been empirically observed that overparameterization helps to improve performance on both the majority and minority subgroups of the data. Few works have also proposed methods to improve the performance on minority subgroups like group distributionally robust optimization and data subsampling. This paper confirms these findings theoretically by studying this using a random features linear regression model which takes into account the difference between the true predictors of the subgroups, the ratio of the data and the signal to noise ratio.

**Summary Of The Review:**

The main concern I have with this paper is the concept shift model that this paper assumes for the subpopulation setting and lack of citations and comparison to previous theoretical works.

---

### Official Review · Reviewer_y1cp · 2022-10-30

**Confidence:** 3
**Correctness:** 3
**Technical Novelty And Significance:** 2
**Empirical Novelty And Significance:** 1
**Recommendation:** 5

**Clarity, Quality, Novelty And Reproducibility:**

The paper is well-written, with a clear exposition of messages and well-designed experiments. I think the choice of models should be better justified.

**Strength And Weaknesses:**

- **Strength**:
The paper is well-written and the simulations are well-designed. The message is clearly conveyed.

- **Major concern**:
On page 3, it is remarked that the paper adopts the random feature model instead of the linear model because results in the latter model do not coincide with the empirical findings. I am confused in that (1) what is the model in general used in empirical works (it seems to me should be the latter)? (2) if both models are approximations of the models used in practice, then why are there inconsistencies? Choosing the model that exhibits the desired performance feels a bit like cherry-picking to me---but I could be wrong!




**Summary Of The Paper:**

The paper studies the effect of overparametrization on minority groups from a theoretical perspective. Building upon existing works, the authors study the asymptotic performance of overparametrized methods in minority groups in a stylized model and show that their results are consistent with empirical observations.

**Summary Of The Review:**

The paper is well-written and the message is clear; more justifications are needed in terms of the choice of model.

---

### Official Review · Reviewer_1vd9 · 2022-11-08

**Confidence:** 2
**Clarity, Quality, Novelty And Reproducibility:** The paper is clear but lacks theoreti…
**Correctness:** 2
**Technical Novelty And Significance:** 2
**Empirical Novelty And Significance:** 1
**Recommendation:** 3

**Strength And Weaknesses:**

The paper is easy to follow and well-written. The numerical experiments are abundant.

However, I find the paper lacks theoretical novelty as the main results are direct applications of previous work. In the abstract the authors claim that they show overparameterization always improves minority group performance, but I found no such theorem is stated instead of numerical solutions to the fixed-point equations characterizing the asymptotic behavior. I would expect a theorem that says the effective bias and variance on the minority group is monotone if it is claimed that this is shown.

**Summary Of The Paper:**

The paper studies the effect of overparameterization under the existence of minority groups. The paper provides theoretical characterizations based on existing work, and ran experiments on the fixed point equations to show that the overparamterization does reflect recent empirical findings in this high dimensional setup.

**Summary Of The Review:**

See above.

---

### Official Review · Reviewer_Fo6C · 2022-11-19

**Confidence:** 4
**Correctness:** 3
**Technical Novelty And Significance:** 2
**Empirical Novelty And Significance:** Not applicable
**Recommendation:** 3

**Clarity, Quality, Novelty And Reproducibility:**

The clarity and the quality of the paper are good. This paper does not have problem of reproducibility. However, this paper lacks enough novelty to meet the bar of ICLR.

**Strength And Weaknesses:**

Strength: This paper is well written, easy to follow, and has solid results.

I think the paper has the following key weakness
The results seem to be very straightforward when compared to under-parametrized cases, but this paper is lacking the comparison of over-parametrized cases and over-parametrized cases, including the clarifications/justifications whether and why the conclusions made for under-parametrized cases can or cannot apply the over-parametrized cases. If the results for over-parametrized cases do not differ from the under-parametrized cases, the novelty and necessity of this paper may be problematic.

Also, while this paper is discussing the over-parametrized ML, its analysis and/or conclusions do not cover how the minority groups' performance change with the number of parameters from under-parametrized cases to over-parametrized cases.

Finally, the two key conclusions drawn from the paper (minority groups perform worse and sub-sampling can help) are pretty intuitive and within the expectation of most people. This paper does not show great novelties of the methods, so the contribution of this paper to the community does not seem high.

**Summary Of The Paper:**

This paper studies the problem of how overparametrization affects the ERM's performance on minority groups theoretically. To be more specific, this paper theoretically shows that under the overparametrazation condition and ERM algorithm, the minority groups will tend to have worse performance on the regression tasks, and also shows that subsampling majority groups can improve the performance of the minority groups. The theoretical results are consistent with the empirical studies observed in previous works.

**Summary Of The Review:**

This paper has three key limitations stated in section "strength and weakness" and lacks the novelty and impacts to the community. So I tend to vote a reject to this paper.

---

### Decision · Program_Chairs · 2023-01-20

**Decision:**

Reject

**Justification For Why Not Higher Score:**

I would recommend rejection for this paper because of the following reasons:

- The main results of this paper lack novelty.
- Lack of comparison to over-parameterized cases and lack of clarity on whether the conclusions apply to such cases, as well as lack of analysis of how minority groups’ performance changes with the number of parameters.
- The claim made in the abstract that overparameterization always improves minority group performance is unsupported by a theorem in the paper.


**Justification For Why Not Lower Score:**

N/A

**Metareview: Summary, Strengths And Weaknesses:**

Summary
---
This paper investigates the impact of overparameterization on the performance of the empirical risk minimization (ERM) algorithm on minority groups. The paper presents theoretical findings that suggest that under overparameterization, minority groups tend to perform worse on regression tasks, but that subsampling majority groups can improve the performance of minority groups. The paper also confirms these findings by studying this using a random features linear regression model that considers the differences in true predictors between subgroups, the ratio of data to signal-to-noise ratio. The results of the paper are consistent with empirical observations in previous research.

Strengths
---
This paper investigates the impact of overparameterization on minority groups in machine learning, which is an important topic for fairness and safety. The paper is easy to follow.

Weaknesses
---
- Lack of comparison to over-parameterized cases and lack of clarity on whether the conclusions apply to such cases, as well as lack of analysis of how minority groups’ performance changes with the number of parameters.
- Lack of a stated theorem to support the claim that overparameterization always improves minority group performance, with only numerical solutions to fixed-point equations being provided to characterize asymptotic behavior
- There are inconsistencies between the models studied in the paper and the practical setting that this work motivates
- Unfounded assumption that subgroups have the same feature distribution and only differ in their true conditional output distribution
- The two key conclusions  (minority groups perform worse and sub-sampling can help)  drawn from the paper being fairly intuitive and not showing great novelty.